# Guaranteed Diversity and Optimality in Cost Function Network Based Computational Protein Design Methods †

**Manon Ruffini** [1], **Jelena Vucinic** [2], **Simon de Givry** [1], **George Katsirelos** [3], **Sophie Barbe** [2] **and Thomas Schiex** [1,*]

1   Université Fédérale de Toulouse, ANITI, INRAE, UR 875, 31326 Toulouse, France; manon.ruffini@inrae.fr (M.R.); simon.de-givry@inrae.fr (S.d.G.)
2   TBI, Université de Toulouse, CNRS, INRAE, INSA, ANITI, 31077 Toulouse, France; jelena.vucinic@inrae.fr (J.V.); sophie.barbe@insa-toulouse.fr (S.B.)
3   MIA-Paris-Mathématiques et Informatique Appliquées, INRAE, 75231 Paris, France; georgios.katsirelos@inrae.fr
*   Correspondence: thomas.schiex@inrae.fr
†   This paper is an extended version of our paper published in the Proceedings of the 2019 IEEE 31st International Conference on Tools with Artificial Intelligence, Portland, OR, USA, 4–6 November 2019.

**Abstract:** Proteins are the main active molecules of life. Although natural proteins play many roles, as enzymes or antibodies for example, there is a need to go beyond the repertoire of natural proteins to produce engineered proteins that precisely meet application requirements, in terms of function, stability, activity or other protein capacities. Computational Protein Design aims at designing new proteins from first principles, using full-atom molecular models. However, the size and complexity of proteins require approximations to make them amenable to energetic optimization queries. These approximations make the design process less reliable, and a provable optimal solution may fail. In practice, expensive libraries of solutions are therefore generated and tested. In this paper, we explore the idea of generating libraries of provably diverse low-energy solutions by extending cost function network algorithms with dedicated automaton-based diversity constraints on a large set of realistic full protein redesign problems. We observe that it is possible to generate provably diverse libraries in reasonable time and that the produced libraries do enhance the Native Sequence Recovery, a traditional measure of design methods reliability.

**Keywords:** computational protein design; graphical models; automata; cost function networks; structural biology; diversity

## 1. Introduction

Proteins are complex molecules that govern much of how cells work, in humans, plants, and microbes. They are made of a succession of simple molecules called *α*-amino acids. All *α*-amino acids share a common constant linear body and a variable sidechain. The sidechain defines the nature of the amino acid. There are 20 natural amino acid types, each with a distinct sidechain offering specific physico-chemical properties. In a protein, the successive amino acids are connected one to the other by peptidic bonds, defining a long linear polymer called the protein backbone. In solution, most proteins fold into a 3D shape, determined by the physico-chemical properties of their amino acid sidechains. Because of their wide variety of functions, and their potentials for applications in medicine, environment, biofuels, green chemistry, etc., new protein sequences are sought that present desired new or enhanced properties and functions. As function is closely related to three-dimensional (3D) structure [1], computational protein design (CPD) methods aim at finding a sequence that folds into a target 3D structure that corresponds to the desired properties and functions. A general formulation of this problem being highly intractable, simplifying assumptions have been made (see Figure 1): the target protein structure (or backbone) is

often assumed to be rigid, the continuous space of flexibility of amino acids sidechains is represented as a discrete set of conformations called rotamers and the atomic forces that control the protein stability are represented as a decomposable energy function, defined as the sum of terms involving at most two bodies (amino acids). The problem of design is then reduced to a purely discrete optimization problem: given a rigid backbone, one must find a combination of discrete sidechain natures and conformations (rotamers) that minimizes the energy. The resulting sequence and associated sidechain conformations define the Global Minimum Energy Conformation (GMEC) for the target backbone. A rotamer library for all 20 natural amino acids containing typically a few hundreds of conformations, the discrete search space becomes very quickly challenging to explore and the problem has been shown to be NP-hard [2] (decision NP-complete). It has been naturally approached by stochastic optimization techniques such as Monte Carlo simulated annealing [3], as in the commonly used Rosetta software [4]. Such stochastic methods offer only asymptotic optimality guarantees. Another possible approach is to use provable optimization techniques that instead offer finite-time deterministic guarantees. In the last decade, Constraint programming-based algorithms for solving the weighted constraint satisfaction problem (WCSP) on cost function networks (CFN) have been proposed to tackle CPD instances [5,6]. These provable methods have shown unprecedented efficiency at optimizing decomposable force fields on genuine protein design instances [6], leading to successfully characterized new proteins [7]. Cost Function Networks are one example of a larger family of mathematical models that aim at representing and analyzing decomposable functions, called graphical models [8,9].

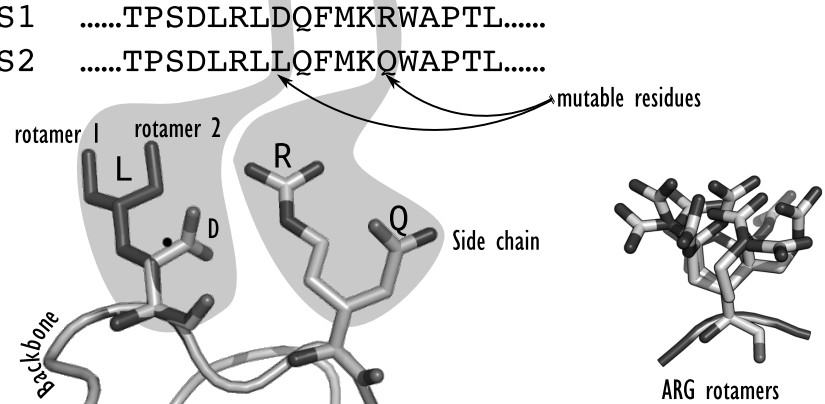

**Figure 1.** An example of two protein sequences (top) where two mutable amino acids have been redesigned. At the first position, the amino acid D (an aspartic acid) has been changed to a L (leucine), in a specific conformation (orientation). At the second position, the arginine R, with its very long and flexible sidechain has been changed to a glutamine Q. The figure on the right illustrates the potential flexibility of the long arginine sidechain, showing a sample of several possible superimposed conformations, representing a fraction of all possible conformations for an arginine sidechain in existing rotamer libraries.

Even if provable methods definitely remove the possibility of failed optimization, they cannot fight the simplifying assumptions that appear in the CPD problem formulation. First, the optimized pairwise decomposed energetic criterion only approximates the actual molecule energy. Then, the rigid backbone and discrete-chain conformations ignore the actual continuous protein flexibility [10]. Ultimately, even with a perfect energy function, an alternative backbone structure may well exist that gives the GMEC sequence an even better energy. This usually requires expensive post hoc filtering based on structure prediction (forward folding [11]). Therefore, even with provable methods, a library of highly relevant mutants is usually produced for further experimental testing, with the hope that the probability of identifying a functional protein will be increased. Provable Branch-and-Bound-based WCSP algorithms have the native ability of enumerating solutions within a

threshold of the optimal solution. Empirically, one can observe that the set of sequences that lie within this threshold grows very quickly in size with the energy threshold, but is mostly composed of sequences that are very similar to the optimal GMEC sequence. Ideally, a design library should be a set of low energy but also diverse solutions. With yeast-display capacity to simultaneously express and test thousands of proteins, libraries of diversified designs become increasingly important. The hope is that sequence diversity will improve the likelihood that a protein endowed of desired function is found. In this paper, we are therefore interested in algorithmic methods that can provide such a set of guaranteed diverse low-energy solutions and then to empirically check if enforcing diversity in a library while optimizing energy does improve the protein design process.

Because of their important applications, protein sequences can be subject to patents. Ideally, a newly designed sequence should satisfy a *minimum* Hamming distance constraint to existing patented sequences. Specific design targets may need to escape known patterns such as e.g., antigenic sub-sequences that would be recognized by the Major Histocompatibility Complexes [12]. This again raises the need to produce sequences satisfying minimum distance requirement to given sequences. Finally, CPD input structures often come from existing, experimentally resolved natural (or native) proteins. In this case, a native sequence exists that has usually acquired desirable properties following the billions of years of natural evolution and optimization it has been through. In many cases, to avoid disrupting the native protein properties (e.g., catalytic capacities), the protein designer may want to bound the maximum number of mutations introduced in the design. This raises the need to produce sequences satisfying *maximum* distance requirement to given sequences.

In this paper, given an initial rigid backbone, we consider the problem of producing a set of diverse low-energy sequences that also provably satisfy a set of minimum and maximum distance requirements regarding given sequences. We observe that beyond structural biology and bioinformatics, this problem of producing a set of diverse solutions of a graphical model has been considered by many authors, either on discrete Boolean graphical models such as constraint networks (used in constraint programming), or on stochastic graphical models such as Markov random fields. Although this shows that the interest for the problem of diverse solutions generation goes well beyond computational protein design, we observe that these approaches either offer no guarantee, or are limited to specific tractable sub-classes of functions, such as submodular functions [13]. Our approach instead relies on the reduction of distance requirements to discrete automaton-based constraints that can be decomposed and compressed into three-bodies (ternary) or two-bodies (binary) terms, using suitable dual and hidden encoding [14,15]. These constraints can then be processed natively by existing WCSP algorithms. Although our approach is general and generally applicable to the production of libraries of solutions of arbitrary discrete graphical models, its design is motivated by computation protein design. We therefore empirically evaluate this approach for the generation of a library of diverse sequences on a set of protein design problems. We first observe that this approach can produce provably diverse sets of solutions on computational protein design problems of realistic sizes in reasonable time. Going back to our initial aim, we also observe that sufficiently diverse libraries do offer better Native Sequence Recovery rates (NSR), a usual metric for protein design methods evaluation that measures how well it is able to reproduce nature's optimization.

## 2. Computational Protein Design

A CPD instance is first composed of an input target 3D structure, defined by the Cartesian coordinates of all the atoms in the protein backbone. The target protein structure can come from an existing protein backbone that was determined experimentally on an existing protein; or from a model that can be derived from existing 3D structures of similar proteins; or from a completely new structure, as it is done in *de novo* design. Once a backbone has been chosen, the design space must be fixed. One may choose to do a full redesign, where the amino acids at all positions of the protein can be changed, or redesign

only a subset of all positions, focusing on positions that are key for the targeted function. Overall, each position in the protein sequence will be set by the designer as either *fixed*, *flexible*, or *mutable*. If the position is *fixed*, the sidechain is fixed and rigid: the amino acid type and orientation are determined in the input target structure. If the position is *flexible*, the residue type is fixed to the amino acid type of the input structure, but the sidechain might adopt several conformations in space. If the position is *mutable*, all or a restricted set of amino acid types are allowed at the position, along with different conformations of their sidechain. Because of the supposed rigidity of the backbone, the sequence-conformation search space is therefore characterized by two decision levels: the sequence space, which corresponds to all the possible sequences **s** enabled by the mutable positions, and the conformation space, which must be searched to identify the best sidechain conformation at each flexible or mutable position. The possible conformations, or rotamers, for each amino acid are indexed in *rotamer libraries*, such as the Dunbrack [16] or the Penultimate libraries [17]. Each library gathers a finite set of conformations, capturing a representative subset of all frequently adopted conformations in experimentally determined structures. In the Rosetta design software [4] that relies on the Dunbrack library, a fully mutable position will be typically associated with around 400 possible rotamers. Designing a 10-residue peptide actually requires the exploration of $400^{10} \approx 10^{26}$ conformations.

Given a backbone structure and a rotamer library, the CPD problem seeks a stable and functional sequence-conformation. The protein functionality is assumed to result from its conformation and its stability is captured by an energy function $E$ that allows computation of the energy of any sequence-conformations on the target backbone. The task at hand is the minimization of this energy function. The optimal sequence is the best possible choice for the target rigid backbone. To model the energy, score functions are used. They can be physics-based, as the energetic force fields AMBER [18] and CHARMM [19]. They capture various atomic interactions including bond and torsion angle potentials, van der Waals potentials, electrostatic interactions, hydrogen bonds forces and entropic solvent effects. Score functions may also be enriched by "knowledge-based" energy terms that result from the statistical analysis of known protein structures. For instance, Rosetta `ref2015` and `beta_nov_16` score functions [4,20] also integrate rotamer log-probabilities of apparition in natural structures, as provided in the Dunbrack library, in a specific energy term. To be optimized, the energy function should be easy to compute while remaining as accurate as possible, to predict relevant sequences. To try to meet these requirements, *additive pairwise decomposable* approximations of the energy have been chosen for protein design approaches [6,21]. The decomposable energy $E$ of a sequence-conformation $\mathbf{r} = (r_1, \dots, r_n)$ where $r_i$ is the rotamer used at the position $i$ in the protein sequence can be written as:

$$E(\mathbf{r}) = E_\varnothing + \sum_{1 \leqslant i \leqslant n} E_i(r_i) + \sum_{1 \leqslant i < j \leqslant n} E_{ij}(r_i, r_j)$$

The term $E_\varnothing$ is a constant that captures interactions within the rigid backbone. For $1 \leqslant i \leqslant n$, the unary (or one body) terms $E_i$ capture the interactions between the rotamer $r_i$ at position $i$ and the backbone, as well as interactions internal to the rotamer $r_i$. For $1 \leqslant i < j \leqslant n$, the binary terms $E_{ij}$ capture the interactions between rotamers $r_i$ and $r_j$ at positions $i$ and $j$ respectively. These energy terms only vary with the rotamers, thanks to the rigid backbone assumption. Protein design dedicated software, such as OSPREY [22] or Rosetta [4], compute all the constant, unary, and binary energy terms, for each rotamer and combination of rotamers, for each position and pair of positions. While this requires quadratic time in the protein length in the worst case, distance cutoffs make these computations essentially linear in this length. Once computed, these values are stored in energy matrices. During the exploration of the sequence-conformation space, conformation energies can be efficiently computed by summing the relevant energy terms fetched from the energy matrix. CPD methods aim at finding the optimum conformation, called the *global minimum energy conformation* (GMEC). Despite all these simplifications, this problem remains decision NP-complete [23].

### 3. CPD as a Weighted Constraint Satisfaction Problem

A cost function network (CFN) $\mathcal{C}$ is a mathematical model that aims at representing functions of many discrete variables that decompose as a sum of simple functions (with small arity or concise representation). It is a member of a larger family of mathematical models called graphical models [9] that all rely on multivariate function decomposability. A CFN is defined as a triple $\mathcal{C} = (\mathbf{X}, \mathbf{D}, \mathbf{C})$ where $\mathbf{X} = (X_1, \ldots, X_n)$ is a set of variables, $\mathbf{D} = (\mathbf{D}_1, \ldots, \mathbf{D}_n)$ is a set of finite domains, and $\mathbf{C}$ is a set of cost functions. Each variable $X_i$ takes its values in the domain $\mathbf{D}_i$. Each cost function $c_\mathbf{S} \in \mathbf{C}$ is a non-negative integer function that depends on the variables in $\mathbf{S}$, called the scope of the function. Given a set of variables $\mathbf{S} \subset \mathbf{X}$, the set $\mathbf{D_S} \prod_{X_i \in \mathbf{S}} \mathbf{D}_i$ denotes the Cartesian product of the domains of the variables in $\mathbf{S}$. For a tuple $\mathbf{t} \in \mathbf{Y}$, with $\mathbf{S} \subset \mathbf{Y} \subset \mathbf{X}$, the tuple $\mathbf{t}[\mathbf{S}]$ denotes the projection of $\mathbf{t}$ over the variables of $\mathbf{S}$. A cost function $c_\mathbf{S} \in \mathbf{C}$ maps tuples of $\mathbf{D_S}$ to integer costs in $\{0, \ldots, \top\}$. In this paper, we assume, as is usual in most graphical models, that the default representation of a cost function $c_\mathbf{S}$ is a multidimensional cost table (or tensor) that contains the cost of every possible assignment of the variables in its scope. This representation requires space that grows exponentially in the cost function arity $|\mathbf{S}|$ which explains why arity is often assumed to be at most two. The joint function is defined as the bounded sum of all cost functions in $\mathbf{C}$:

$$C_\mathcal{C} : \quad \begin{aligned} \mathbf{D_X} &\longrightarrow \{0, \ldots, \top\} \\ \mathbf{t} &\longmapsto \sum^{\top}{}_{c_\mathbf{S} \in \mathbf{C}} c_\mathbf{S}(\mathbf{t}[\mathbf{S}]) \end{aligned}$$

where the bounded sum $+^\top$ is defined with $a +^\top b = \min(a + b, \top)$. The maximum cost $\top \in \mathbb{N} \cup \{+\infty\}$ is used for forbidden partial assignments and represents a sort of infinite or unbearable cost. Cost functions that take their values in $\{0, \top\}$ represent hard constraints. The weighted constraint satisfaction problem consists of finding the assignment of all the variables in $\mathbf{X}$ with minimum cost:

$$\mathbf{t}^* = \min_{\mathbf{t} \in \mathbf{D_X}} C_\mathcal{C}(\mathbf{t}) = \min_{\mathbf{t} \in \mathbf{D_X}} \sum_{c_\mathbf{S} \in \mathbf{C}} c_\mathbf{S}(\mathbf{t}[\mathbf{S}])$$

Notice that when the maximum cost $\top = 1$, the cost function network becomes a constraint network [24], where cost functions encode only constraints. Tuples that are assigned cost 0 are valid, i.e., they satisfy the constraint, and tuples that are assigned cost $1 = \top$ are forbidden. The Constraint Satisfaction Problem then consists of finding a satisfying assignment, one that satisfies all the constraints.

Graphical models also encompass stochastic networks, such as discrete Markov random fields (MRF) and Bayesian Nets [25]. A discrete *Markov Random Field* is a graphical model $\mathcal{M} = (\mathbf{X}, \mathbf{D}, \mathbf{\Phi})$ where $\mathbf{X} = (X_1, \ldots, X_n)$ is a set of random variables, $\mathbf{D} = (\mathbf{D}_1, \ldots, \mathbf{D}_n)$ is a set of finite domains, and $\mathbf{\Phi}$ is a set of potential functions. A potential function $\varphi_\mathbf{S}$ maps $\mathbf{D_S}$ to $[0, +\infty]$. The *joint potential function* is defined as:

$$P = \Phi_\mathcal{M} : \quad \begin{aligned} \mathbf{D_X} &\longrightarrow [0, +\infty] \\ \mathbf{t} &\longmapsto \prod_{\varphi_\mathbf{S} \in \mathbf{\Phi}} \varphi_\mathbf{S}(\mathbf{t}[\mathbf{S}]) \end{aligned}$$

Instead of the sum used to combine functions in cost function networks, the product is used here. The normalization of the potential function $P$ by the partition function $Z = \sum_{\mathbf{t} \in \mathbf{D_X}} P(\mathbf{t})$ defines the probability function $p = \frac{1}{Z} P$ of the MRF. The Maximum A Posteriori probability corresponds to the assignment with maximum probability (and maximum potential) $\max_\mathbf{t} p(\mathbf{t})$.

MRF can also be expressed using additive energy functions $e_\mathbf{S} \in \mathbf{E}$, a logarithmic transformation $e_\mathbf{S} = -\log \varphi_\mathbf{S}$ of the potential functions. The potential function is then an exponential of the energy $P(\mathbf{t}) = \exp(-\sum_{e_\mathbf{S} \in \mathbf{E}} e_\mathbf{S})$. Although potential functions are multiplied together, energies simply add up. Therefore, cost function networks are closely related to the energetic expression of Markov random fields. The main difference lies in the fact that CFNs deal with non-negative integers only, whereas MRF energies are real

numbers. If $\top = +\infty$, a CFN can be transformed into an MRF through an exponential transformation, and given a precision factor, an MRF can be transformed into a CFN through a log transform. Zero potentials are mapped to cost $\top$ and minimum energy means maximum probability.

Given a cost function network, the weighted constraint satisfaction problem can be answered by the exploration of a search tree in which nodes are CFNs induced by *conditioning*, i.e., domain reductions on the initial network. A *Branch-and-Bound* strategy is used to explore the search tree, which relies on a *lower bound* on the optimum assignment cost in the current subtree. If the lower bound is higher than the best joint cost found so far, it means that no better solution is to be found in the subtree, and it can be pruned. Each time a new solution is found, the maximum cost $\top$ is updated to the corresponding cost, as we are only interested in finding solutions with lower cost. Ordering strategies are crucial and can lead to huge improvement in the empirical computation time: decisions should be made that lead to low-cost solutions (and decrease the maximum cost $\top$) and that enable early pruning.

The efficiency of the branch-and-bound strategy relies on strength of the lower bound on solution costs. In CFNs, since cost functions are non-negative, the empty-scoped cost function $c_\varnothing$ provides a naive lower bound on the optimum cost. To efficiently compute tight lower bounds, local reasoning strategies are used that aim at pushing as much cost as possible in the constant cost $c_\varnothing$, for better pruning. They are based on *equivalence preserving transformations* that perform cost transfers between cost functions, while maintaining the solutions' joint costs unchanged [26], i.e., the joint cost function is preserved (these operations are called reparameterizations in MRFs). Specific sequences of equivalence preserving transformations can be applied to a CFN to improve the lower bound $c_\varnothing$ until a specific target property is reached on the CFN. These properties, called *local consistencies*, aim at creating zero costs, while improving $c_\varnothing$. These sequences of operations should converge to a fixpoint (closure). Various local consistency properties have been introduced, as node consistency, arc consistency, full directional arc consistency, existential directional arc consistency, or virtual arc consistency [26]. For binary CFNs that involve functions of arity at most two, these properties can be enforced in polynomial time in the size of the network. Among these, Virtual arc consistency has been shown to solve the WCSP on networks composed of submodular functions [27]. Note however that the complexity of local consistency enforcing remains exponential in the arity of the cost functions (as is the size of the corresponding tensors).

Sometimes, one may need to include specific functions with a large scope in the description of the joint function. Because of the exponential size of the tensor description, these *global cost functions* must be represented with dedicated concise descriptions and require dedicated algorithms for local consistency propagation. More formally, a global cost function, denoted $GCF(\mathbf{S}, \mathcal{A})$, is a family of cost functions, with scope $\mathbf{S}$ and possible parameters $\mathcal{A}$. A global cost function is said to be *tractable* when its minimum can be computed in polynomial time.

The CFN formulation of computational protein design is straightforward [5,6,28,29] (see Figure 2): given a CPD instance with pairwise decomposable energy function $E = E_\varnothing + \sum_{1 \leqslant i \leqslant n} E_i + \sum_{1 \leqslant i < j \leqslant n} E_{ij}$, let $\mathcal{C} = (\mathbf{X}, \mathbf{D}, \mathbf{C})$ be a cost function network with variables $\mathbf{X} = (X_1, \ldots, X_n)$, with one variable $X_i$ for each flexible or mutable position $i$ in the protein sequence, domains $\mathbf{D} = (\mathbf{D}_1, \ldots, \mathbf{D}_n)$ where the domain $\mathbf{D}_i$ of the variable $X_i$ consists of the available rotamers at position $i$, i.e., the amino acid types and their sidechain conformations. The cost functions are the empty-scoped, unary and binary energy terms:

$$\mathbf{C} = \{E_\varnothing\} \cup \{E_i, 1 \leqslant i \leqslant n\} \cup \{E_{ij}, 1 \leqslant i < j \leqslant n\}$$

Energy terms, which are floating-point numbers, are transformed into non-negative integer values by being shifted by a constant, multiplied by a large precision factor, and having their residual decimal truncated.

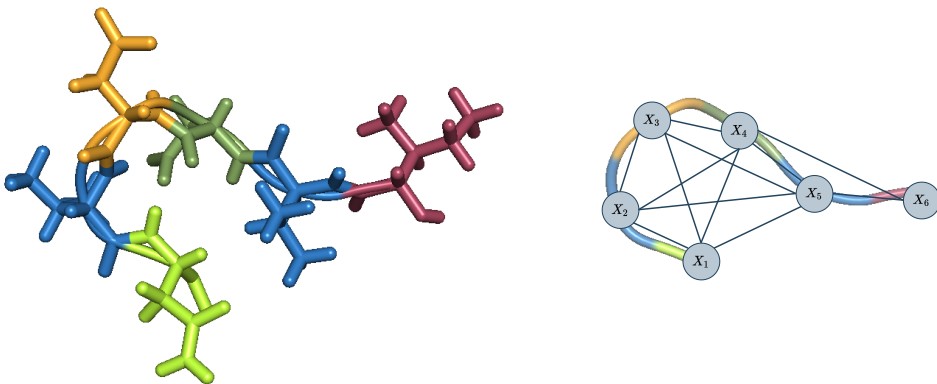

**Figure 2.** Input backbone and cost function network representation of a corresponding CPD instance with 6 mutable or flexible residues.

In this encoding, variables represent rotamers, combining information on the nature and the geometry (conformation) of the sidechain. In practice, it is often useful to add extra variables that represent the sequence information alone. The CFN $\mathcal{C}$ can be transformed into $\mathcal{C}' = (\mathbf{X}', \mathbf{D}', \mathcal{C}')$, that embeds sequence variables, as follows:

**Variables** We add sequence variables to the network: $\mathbf{X}' = \mathbf{X}^{seq} \cup \mathbf{X}$, where $\mathbf{X}^{seq} = \{X_i^{seq} | X_i \in \mathbf{X}\}$. The value of $X_i^{seq}$ represents the amino acid type of the rotamer value of $X_i$.

**Domains** $\mathbf{D} = \mathbf{D}^{seq} \cup \mathbf{D}$ where $\mathbf{D}^{seq} = \{\mathbf{D}_i^{seq} | \mathbf{D}_i \in \mathbf{D}\}$ where the domain $\mathbf{D}_i^{seq}$ of $X_i^{seq}$ is the set of available amino acid types at position $i$.

**Constraints** The new set of cost functions $\mathbf{C}'$ is made of the initial functions $\mathbf{C}$; and sequence constraints that ensure that $X_i^{seq}$ is the amino acid type of rotamer $X_i$. Such a function $c_{X_i, X_i^{seq}}$ just forbids (map to cost $\top$) pairs of values $(r, a)$ where the amino acid identity of rotamer $r$ does not match $a$. All other pairs are mapped to cost $0$.

Such sequence variables depend functionally on the rotamer variables. They do not modify the search space and merely offer a facility to define properties on the protein sequence, if needed, as will be the case here.

## 4. Diversity and Optimality

In this section, we assume that we have a CFN $\mathcal{C} = (\mathbf{X}, \mathbf{D}, \mathbf{C})$ and we want to express diversity requirements on the values taken by variables in $\mathbf{S} \subset \mathbf{X}$. In the case of CPD, these variables will be the previously introduced *sequence* variables.

### 4.1. Measuring Diversity

The task at hand is the production of a *set* of *diverse* and *low-cost* solutions of $\mathcal{C}$. First, we need a measure of diversity between pairs of solutions, and among sets of solutions.

The simplest diversity measure between pairs of solutions is the *Hamming distance*, defined hereafter. It counts the number of variables in $\mathbf{S}$ that take different values in two solutions. In the CPD framework, sequence variables represent amino acid identities: the Hamming distance measures the number of introduced mutations (or substitutions), a very usual notion in protein engineering.

**Definition 1.** *Given a set of variables* $\mathbf{S} \subset \mathbf{X}$ *and two assignments* $\mathbf{t}$ *and* $\mathbf{t}' \in \mathbf{D_S}$, *the* Hamming distance *between* $\mathbf{t}$ *and* $\mathbf{t}'$ *is defined as follows:*

$$d_H(\mathbf{t}, \mathbf{t}') = \sum_{X_i \in \mathbf{S}} \mathbb{1}(\mathbf{t}[X_i] \neq \mathbf{t}'[X_i])$$

The Hamming distance can be generalized to take into account dissimilarity scores between values. The resulting distance is a semi-metric, defined as a sum of variable-wise dissimilarities, as follows:

**Definition 2.** *Given a zero-diagonal symmetric positive matrix D that defines value dissimilarities, and two assignments* $\mathbf{t}, \mathbf{t}' \in \mathbf{D_S}$, *the* weighted-Hamming distance *between* $\mathbf{t}$ *and* $\mathbf{t}'$ *is defined as:*

$$d_D(\mathbf{t}, \mathbf{t}') = \sum_{X_i \in \mathbf{S}} D(\mathbf{t}[X_i], \mathbf{t}'[X_i])$$

In computational biology, protein sequences are often compared using dedicated similarity matrices, such as the BLOSUM62 matrix [30]. A protein similarity matrix $S$ can be transformed into a dissimilarity matrix $D$ such that $D_{i,j} = (S_{i,i} + S_{j,j})/2 - S_{i,j}$.

**Definition 3.** *Given a set* $\mathbf{Z}$ *of solutions, we define:*

- *its* average dissimilarity: $\bar{d}(\mathbf{Z}) = \dfrac{2}{|\mathbf{Z}|(|\mathbf{Z}|-1)} \sum_{\mathbf{t} \neq \mathbf{t}' \in \mathbf{Z}} d(\mathbf{t}, \mathbf{t}')$

- *its* minimum dissimilarity: $\check{d}(\mathbf{Z}) = \min_{\mathbf{t} \neq \mathbf{t}' \in \mathbf{Z}} d(\mathbf{t}, \mathbf{t}')$

We are aiming at producing a library of solutions that is guaranteed to be diverse. The *average dissimilarity* does not match this need: a set of solutions might have a satisfying average dissimilarity value, with several occurrences of the same assignment, and one or a few very dissimilar ones. Therefore, to guarantee diversity, the *minimum dissimilarity* will be the diversity measure used throughout this paper.

Therefore, producing a set of *diverse* solutions requires that all solution pairs have their distance above a given threshold. This can be encoded in cost functions representing constraints, taking their values in $\{0, \top\}$ only:

**Definition 4.** *Given two sets of variables* $\mathbf{S}, \mathbf{S}'$ *of the same cardinality, a dissimilarity matrix D and a diversity threshold* $\delta$, *we define the global cost function:*

$$\mathrm{DIST}(\mathbf{S}, \mathbf{S}', D, \delta): \quad \begin{aligned} \mathbf{D_S} \times \mathbf{D_{S'}} &\longrightarrow \{0, \top\} \\ (\mathbf{t}, \mathbf{t}') &\longmapsto \begin{cases} 0 & \text{if } sign(\delta).d(\mathbf{t}, \mathbf{t}') \geqslant \delta \\ \top & \text{otherwise.} \end{cases} \end{aligned}$$

Allowing both positive and negative threshold $\delta$ allows the DIST cost function to express either minimum or maximum diversity constraints. When $\delta > 0$, the cost function expresses a minimum dissimilarity requirement between the assignments $\mathbf{t}$ and $\mathbf{t}'$:

$$\mathrm{DIST}(\mathbf{t}, \mathbf{t}', D, \delta) = 0 \Leftrightarrow d(\mathbf{t}, \mathbf{t}') \geqslant \delta$$

If $\delta < 0$, the cost function represents the fact that $\mathbf{t}$ and $\mathbf{t}'$ must be similar, with a dissimilarity lower than the absolute value of $\delta$:

$$\mathrm{DIST}(\mathbf{t}, \mathbf{t}', D, \delta) = 0 \Leftrightarrow -d(\mathbf{t}, \mathbf{t}') \geqslant \delta \Leftrightarrow d(\mathbf{t}, \mathbf{t}') \leqslant -\delta = |\delta|$$

If needed, both maximum and minimum requirements can be imposed using two constraints.

### 4.2. Diversity Given Sequences of Interest

In the CPD context, minimum and maximum distance requirements with known sequences may be useful in practice in at least two situations.

- A native functional sequence $\mathbf{s}_{nat}$ is known for the target backbone. The designer wants that less than $\delta_{nat}$ mutations be introduced on some sensitive region of the native protein, to avoid disrupting a crucial protein property.
- A patented sequence $\mathbf{s}_{pat}$ exists for the same function, and sequences with more than $\delta_{pat}$ mutations are required for the designed sequence to be usable without requiring a license.

The distance here is the Hamming distance based on matrix $H$ which equals 1 everywhere except for its zero diagonal. Using sequence variables, the following diversity constraint-encoding cost functions need to be added to the CFN model:

- $\text{DIST}(\mathbf{X}^{seq}, \mathbf{s}_{nat}, H, -\delta_{nat})$
- $\text{DIST}(\mathbf{X}^{seq}, \mathbf{s}_{pat}, H, \delta_{pat})$

### 4.3. Sets of Diverse and Good Quality Solutions

The problem of producing a set of diverse and good quality solutions, i.e., such that all pairs of solutions satisfy the diversity constraint, and the solutions have minimum cost, can be expressed as follows:

**Definition 5** (DIVERSESET ). *Given a dissimilarity matrix $D$, an integer $M$ and a dissimilarity threshold $\delta$, the problem $\text{DIVERSESET}(\mathcal{C}, D, M, \delta)$ consists of producing a set $\mathbf{Z}$ of $M$ solutions of $\mathcal{C}$ such that:*

**Diversity** *For all $\mathbf{t} \neq \mathbf{t}' \in \mathbf{Z}$, $d(\mathbf{t}, \mathbf{t}') \geqslant \delta$, i.e., $\text{DIST}(\mathbf{t}, \mathbf{t}', D, \delta) = 0$.*

**Quality** *The solutions have minimum cost, i.e., $\sum_{\mathbf{t} \in \mathbf{Z}}^{\top} C_{\mathcal{C}}(\mathbf{t})$ is minimum.*

For a CFN $\mathcal{C}$ with $n$ variables, solving DIVERSESET requires simultaneously deciding the value of $nM$ variables. It can be solved by making $M$ copies of $\mathcal{C}$ with variable sets $\mathbf{X}^1$ to $\mathbf{X}^M$ and adding $\dfrac{M.(M-1)}{2}$ constraints $\text{DIST}(X^i, X^j, D, \delta)$ for all $1 \leqslant i < j \leqslant M$. If the upper bound $\top$ is finite, all its occurrences must be replaced by $M.(\top - 1) + 1$. Although very elegant, this approach yields a CFN instance where the number of variables is multiplied by the number of wanted solutions. The WCSP and CPD problems being NP-hard, one can expect that the resulting model will be quickly challenging to solve. We empirically confirmed this on very tiny instances: we tested it on problems with 20 variables and maximum domain size bounded by six, asking for just for four 15-diverse solutions. This elegant approach took more than 23 h to produce 4 solutions. We therefore decided to solve a relaxed version of DIVERSESET : an iterative algorithm provides a greedy approximation of the problem that preserves most of the required guarantees. Using this approach, the problem of producing four 15-diverse solutions of the above tiny problem takes just 0.28 s.

**Definition 6** (DIVERSESEQ ). *Given a dissimilarity matrix $D$, an integer $M$ and a dissimilarity threshold $\delta$, the set of assignments $\mathbf{Z}$ of $\text{DIVERSESEQ}(\mathcal{C}, D, M, \delta)$ is built recursively:*

- *The first solution $\mathbf{Z}[1]$ is the optimum of $\mathcal{C}$*
- *When solutions $\mathbf{Z}[1..(i-1)]$ are computed, $\mathbf{Z}[i]$ is such that:*
  *for all $1 \leqslant j < i$, $\text{DIST}(\mathbf{Z}[i], \mathbf{Z}[j], D, \delta) = 0$ and $\mathbf{Z}[i]$ has minimum cost.*
  *That is, $\mathbf{Z}[i]$ is the minimum cost solution, among assignments that are at distance at least $\delta$ from all the previously computed solutions.*

The set of solutions DIVERSESEQ needs to optimally assign $n$ variables $M$ times, instead of the $n.M$ variables. Given the NP-hardness of the WCSP, solving DIVERSESEQ

may be exponentially faster than DIVERSESET while still providing guarantees that distance constraints are satisfied together with a weakened form of optimality, conditional on the solutions found in previous iterations. The solution set is still guaranteed to contain the GMEC (the first solution produced).

## 5. Relation with Existing Work

In the case of Boolean functions ($\top = 1$), the work of [31] considers the optimization of the solution set cardinality $M$ or the diversity threshold $\delta$ using average or minimum dissimilarity. The authors prove that enforcing arc consistency on a constraint requiring sufficient average dissimilarity $\bar{d}$ is polynomial but NP-complete for minimum dissimilarity $\check{d}$. They evaluate an algorithm for incremental production of a set maximizing $\bar{d}$. The papers [32,33] later addressed the same problems using global constraints and knowledge compilation techniques. Being purely Boolean, these approaches cannot directly capture cost (or energy) which is crucial for CPD. More recently, ref. [34] proposed a Constraint Optimization Problem approach to provide diverse high-quality solutions. Their approach however trades diversity for quality while diversity is a requirement in our case.

The idea of producing diverse solutions has also been explored in the more closely related area of discrete stochastic graphical models (such as Markov random fields). In the paper of Batra et al. [35], the Lagrangian relaxation of minimum dissimilarity constraints is shown to add only unary cost functions to the model. This approach can be adapted to cost function networks, but a non-zero duality gap remains and ultimately, no guarantee can be offered. This work was extended in [36] using higher-order functions to *approximately* optimize a trade-off between diversity and quality. More recently, ref. [37] addressed the DIVERSESET problem, but using optimization techniques that either provide no guarantee or are restricted to tractable variants of the WCSP problem, defined by submodular functions [13].

In the end, we observe that none of these approaches simultaneously provides guarantees on quality and diversity. Closest to our target, ref. [38] considered the problem of incrementally producing the set of the best $M$ $\delta$-modes of the joint distribution $J_N(X)$.

**Definition 7** ([39])**.** *A solution* **t** *is said to be a $\delta$-mode iff there exists no better solution than* **t** *in the Hamming ball of radius $\delta$ centered in* **t** *(implying that* **t** *is a local minimum).*

In [38–41], an exact dynamic programming algorithm, combined with an A* heuristic search and tree-decomposition was proposed to exactly solve this problem with the Hamming distance. This algorithm relies however on NP-complete lower bounds and is restricted to a fixed variable order, a restriction that is known to often drastically hamper solving efficiency. It however provides a diversity guarantee: indeed, a $\delta$-mode will always be *strictly* more than $\delta$ away from another one and will be produced by greedily solving DIVERSESEQ.

**Theorem 1.** *Given a cost function network $\mathcal{C}$, a diversity threshold $\delta$, and $D = H$ the Hamming dissimilarity matrix, for any $\delta$-mode* **t***, there exists a value $M'$ such that the solution of* DIVERSESEQ$(\mathcal{C}, H, M', \delta + 1)$ *contains* **t***.*

**Proof.** If a $\delta$-mode **t** is not in the solution of DIVERSESEQ$(\mathcal{C}, H, M', \delta + 1)$ , this must be because it is forbidden by a DIST constraint. Consider the iteration $i$ which forbids **t** for the first time: a solution with a cost lower than the cost of **t** was produced (else **t** would have been produced instead) but this solution is strictly less than $\delta + 1$ away from **t** (since **t** is forbidden). However, this contradicts the fact that **t** is a $\delta$-mode.　□

For a sufficiently large $M$, the sequence **Z** of DIVERSESEQ$(N, H, M, \delta + 1)$ solutions will therefore contain all $\delta$-modes and possibly some extra solutions. Interestingly, it is not difficult to separate modes from non-modes.

**Theorem 2.**

1. *Any assignment* **t** *of a CFN* $\mathcal{C} = (\mathbf{X}, \mathbf{D}, \mathbf{C})$ *is a δ-mode iff it is an optimal solution of the CFN* $(\mathbf{X}, \mathbf{D}, \mathbf{C} \cup \{\mathrm{DIST}(\mathbf{X}, \mathbf{t}, H, -\delta)\})$
2. *For bounded δ, this problem is in P.*

**Proof.**

1. The function $\mathrm{DIST}(\mathbf{X}, \mathbf{t}, H, -\delta)$ restricts $\mathbf{X}$ to be within $\delta$ of $\mathbf{t}$. If $\mathbf{t}$ is an optimal solution of $(\mathbf{X}, \mathbf{C} \cup \{\mathrm{DIST}(\mathbf{X}, \mathbf{t}, H, -\delta)\})$ then there is no better assignment than $\mathbf{t}$ in the $\delta$-radius Hamming ball and $\mathbf{t}$ is a $\delta$-mode.
2. For bounded $\delta$, a CFN with $n$ variables and at most $d$ values in each domain, there is $O((nd)^{\delta})$ tuples within the Hamming ball, because from $\mathbf{t}$, we can pick any variable ($n$ choices) and change its value ($d$ choices), $\delta$ times. Therefore, the problem of checking if $\mathbf{t}$ is optimal is in P.

□

## 6. Representing the Diversity Constraint

The key to guaranteeing quality and diversity of solutions in a cost function network is the dissimilarity cost function DIST. Given a complete assignment $\mathbf{t}$, a dissimilarity $D$ and a threshold $\delta$, we need to concisely encode the global diversity constraint $\mathrm{DIST}(\mathbf{X}, \mathbf{t}, D, \delta)$.

### 6.1. Using Automata

Given a solution $\mathbf{t}$, a dissimilarity matrix $D$ and a diversity threshold $\delta > 0$, the cost function $\mathrm{DIST}(\cdot, \mathbf{t}, D, \delta)$ needs to be added to the cost function network. Please note that the function may involve all the network variables: it is a *global cost function* and its representation as one huge, exponential size tensor is not possible.

To encode this function concisely, we exploit the fact that the set of authorized tuples defines a regular language that can be encoded into a finite state automaton and then decomposed in ternary functions [42,43]. Here, we use a weighted automaton to define the weighted regular language of all tuples with their associated cost. A weighted automaton $\mathcal{A} = (\Sigma, \mathbf{Q}, \Delta, Q_0, \mathbf{F})$ encoding $\mathrm{DIST}(\mathbf{X}, \mathbf{t}, D, \delta)$ can be defined as follows:

- The alphabet is the set of possible values, i.e., the union of the variable domains $\Sigma = \bigcup_{i=1}^{n} \mathbf{D}_i$
- The set of states $\mathbf{Q}$ gathers $(\delta + 1) \cdot (n + 1)$ states denoted $q_i^d$:

$$\mathbf{Q} = \{q_i^d \mid 0 \leqslant i \leqslant n, 0 \leqslant d \leqslant \delta\}$$

that represent the fact that the first $i$ values of $\mathbf{X}$ have distance $d$ to the first $i$ values of $\mathbf{t}$. For $d = \delta$, automaton state $q_i^{\delta}$ represents the fact that the first $i$ values of $\mathbf{X}$ have distance $\geqslant \delta$ to the first $i$ values of $\mathbf{t}$.

- In the initial state, no value of $\mathbf{X}$ has been read, and the dissimilarity is 0:

$$Q_0 = q_0^0$$

- The assignment is accepted if it has dissimilarity from $\mathbf{t}$ higher than the threshold $\delta$, hence the accepting state:

$$\mathbf{F} = \{q_n^{\delta}\}$$

- For every value $r$ of $X_i$, the transition function $\Delta : \mathbf{Q} \times \Sigma \times \mathbf{Q}$ defines a 0-cost transition from $q_i^d$ to $q_{i+1}^{\min(d+D(r,\mathbf{t}[i+1]),\delta)}$. All other transitions have infinite cost $\top$.

This weighted automaton contains $O(n \cdot (\delta + 1) \cdot d)$ finite cost transitions, were $d$ is the maximum domain size. An assignment $\mathbf{t}'$ of $\mathbf{X}$ is accepted if and only if $\mathsf{d}(\mathbf{t}', \mathbf{t}) \geqslant \delta$; and the automaton represents the DIST cost function. An example of a DIST encoding automaton is given in Figure 3.

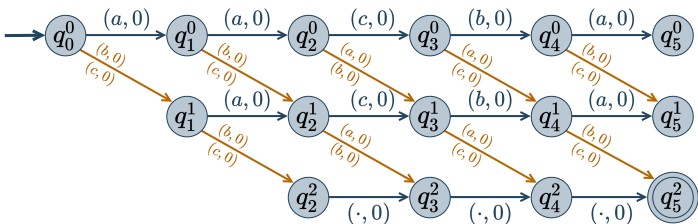

**Figure 3.** Weighted automaton representing $\text{D{\sc ist}}(\mathbf{X}, \mathbf{t}, H, \delta)$ where $\mathbf{X}$ is a set of 5 variables, with domains $\mathbf{D}_i = \{a, b, c\}$, $\mathbf{t} = aacba$, $H$ represents the Hamming distance, and $\delta$ is set to 2. State $q_i^d$ means that values $X_1 \ldots X_i$ are such that $H(X_1 \ldots X_i, t[X_1 \ldots X_i]) = d$ (or $\geqslant \delta$ if $d = \delta$). A labeled arrow $q \xrightarrow{(v,w)} q'$ means $\Delta(q, v, q') = w$, i.e., there is a transition from $q$ to $q'$ with value $v$ and weight $w$.

### 6.2. Exploiting Automaton Function Decomposition

It is known that the W{\sc regular} cost function, encoding automaton $\mathcal{A}$, can be decomposed into a sequence of ternary cost functions [43]. The decomposition is achieved by adding $n + 1$ state variables $Q_0, \ldots, Q_n$, and $n$ ternary cost functions $w^{\mathcal{A}}_{Q_i, X_{i+1}, Q_{i+1}}$, such that $w^{\mathcal{A}}_{Q_i, X_{i+1}, Q_{i+1}}(q_i, x_{i+1}, q_{i+1}) = c$ if and only if there exists a transition from $q_i$ to $q_{i+1}$ in $\mathcal{A}$ labeled with value $x_{i+1}$ and cost $c$. Variable $Q_0$ is restricted to the starting states and variable $Q_n$ to the accepting states. Additional variables and ternary functions are represented in Figure 4. The resulting set of ternary functions is logically equivalent to the original global constraint.

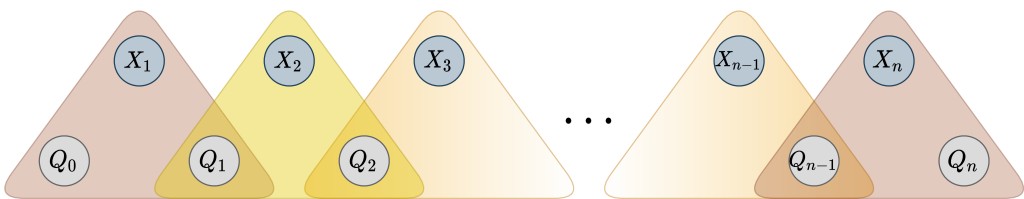

**Figure 4.** Hypergraph representation of the decomposition of a W{\sc regular} cost function with additional state variables $Q_i$ and transition-encoding ternary functions.

The D{\sc ist} function satisfies however several properties that can be exploited to further improve its space and time efficiency. One can first notice that the set of forbidden solutions does not depend on the order of the variables (the distance measure is the sum of independent variable-wise dissimilarities). Therefore, the order of the variables in the automaton can be chosen freely. We can use the DAC-ordering [43]. This order is known to preserve the strength of the lower bounds produced by CFN local consistencies when applied to the decomposition (instead of the initial full D{\sc ist} function, with its potentially exponential size table).

Then, in the case of the D{\sc ist} cost function, each state variable has $(\delta + 1) \cdot (n + 1)$ values (the number of states in the automaton) and each ternary function cost table describes costs of $(\delta + 1)^2 \cdot (n + 1)^2 \cdot d$ tuples, where $d$ is the domain size. To speed up the resolution through better soft local consistency enforcing, we exploit the properties of D{\sc ist} and the dissimilarity matrix $D$.

### 6.3. Compressing the Encoding

The encoding of a D{\sc ist} cost function in a sequence of $n$ ternary functions, described in cost tables of size $(\delta + 1)^2 \cdot (n + 1)^2 \cdot d$ can be reduced along several lines.

First, for D{\sc ist}, we know that states $s_i^d$ can only be reached after $i$ transitions, i.e., the reachable states of variable $Q_i$ are the states in the $i$-th column in the D{\sc ist} automaton (see Figure 3). The domains of the variables $Q_i$ can be reduced to the $\delta + 1$ states $s_i^d$:

$$\mathbf{D}_{Q_i} = \{q_i^d \mid 0 \leqslant d \leqslant \delta\}$$

Furthermore, our semi-metrics are defined by a non-decreasing sum of non-negative elements of $D$. Therefore, any state $q_i^d$ can reach the accepting state $q_n^\delta$ if and only if the maximum dissimilarity $md_i$ that can be achieved from variable $i$ to variable $n$ is larger than the remaining diversity to reach $\delta - d$. All such maximum dissimilarities $md_i$ can be pre-computed in one pass over all variables in $\mathbf{X}$ as follows:

- $md_n = 0$
- For $0 \leqslant i < n$, $md_i = md_{i+1} + max_{v,v' \in \mathbf{D}_{i+1}} D(v, v')$

In the Hamming case, the distance can increase by 1 at most, i.e., $max_{v,v' \in \mathbf{D}_{i+1}} H(v, v') = 1$, therefore $md_i = n - i$.

A symmetric argument holds for the starting state $q_0^0$. These simplifications reduce ternary cost tables to $O((\delta + 1)^2 \cdot d)$.

For a given dissimilarity matrix $D$, let $\#D$ denote the *number of distinct values* that appear in $D$. If variables have domains of maximum size $d$ and ignoring the useless 0 matrix, we know that $2 \leqslant \#D \leqslant 1 + \frac{d \cdot (d+1)}{2}$. However, distance matrices are usually more structured. For example, the BLOSUM62 similarity matrix leads to $\#D = 12$ levels.

In the Hamming case, there are $\#H = 2$ dissimilarity levels. This means that a state $q_i^d$ can only reach states $q_{i+1}^d$ or $q_{i+1}^{d+1}$. This sparsity of the transition matrix can be exploited, provided it is made visible. This can be achieved using extended variants of the *dual* and *hidden* encoding of constraint networks [14,15]. These transformations, detailed hereafter, are known to preserve the set of solutions and their costs.

In constraint networks, the *dual* representation of a constraint network $\mathcal{X} = (\mathbf{X}, \mathbf{D}, \mathbf{C})$ is a new network $\mathcal{X}' = (\mathbf{X}', \mathbf{D}', \mathbf{C}')$ with:

- One variable $X_\mathbf{S}$ per constraint $c_\mathbf{S} \in \mathbf{C}$:

$$\mathbf{X}' = \{X_\mathbf{S} | c_\mathbf{S} \in \mathbf{C}\}$$

- Domain $\mathbf{D}_{X_\mathbf{S}}$ of variable $X_\mathbf{S}$ is the set of tuples $\mathbf{t} \in \mathbf{D}_\mathbf{S}$ that satisfy the constraint $c_\mathbf{S}$:

$$\mathbf{D}' = \{\mathbf{D}_{X_\mathbf{S}} | c_\mathbf{S} \in \mathbf{C}\} \qquad \mathbf{D}_{X_\mathbf{S}} = \{\mathbf{t} \in c_\mathbf{S}\}$$

- For each pair of constraints $c_\mathbf{S}, c_{\mathbf{S}'} \in \mathbf{C}$ with overlapping scopes $\mathbf{S} \cap \mathbf{S}' \neq \varnothing$, there is a constraint $c_{X_\mathbf{S}, X_{\mathbf{S}'}}$ that ensures that tuples assigned to $X_\mathbf{S}$ and $X_{\mathbf{S}'}$ are compatible, i.e., they have the same values on the overlapping variables:

$$\mathbf{C}' = \{c_{X_\mathbf{S}, X_{\mathbf{S}'}} | X_\mathbf{S}, X_{\mathbf{S}'} \in \mathbf{X}', \mathbf{S} \cap \mathbf{S}' \neq \varnothing\}$$

where

$$c_{X_\mathbf{S}, X_{\mathbf{S}'}} = \{(\mathbf{t}, \mathbf{t}') \in \mathbf{D}_{X_\mathbf{S}} \times \mathbf{D}_{X_{\mathbf{S}'}} | \mathbf{t}[\mathbf{S} \cap \mathbf{S}'] = \mathbf{t}'[\mathbf{S} \cap \mathbf{S}']\}$$

We apply this transformation to the reduced $w_{Q_i, X_{i+1}, Q_{i+1}}^{\mathcal{A}}$ functions (see Figure 5). The dual variable of $w_{Q_i, X_{i+1}, Q_{i+1}}^{\mathcal{A}}$ is a variable $X_i^{\mathcal{A}}$ that contains all pairs $(q, q') \in Q_i \times Q_{i+1}$ such that there is a transition from $q$ to $q'$ in $\mathcal{A}$. For the Hamming case, the variable $X_i^{\mathcal{A}}$ has at most $2\delta + 1$ values. It is connected to $X_i$ by a pairwise function:

$$c_{X_i^{\mathcal{A}}, X_i} : \quad \begin{aligned} \mathbf{D}_{X_i^{\mathcal{A}} \times \mathbf{D}_i} &\longrightarrow \{0, \dots, \top\} \\ ((q, q'), v) &\longmapsto \Delta(q, v, q') \end{aligned}$$

where $\Delta$ is the weighted transition function of the automaton $\mathcal{A}$.

In this new dual representation, for every pair of consecutive dual variables $X_{i-1}^{\mathcal{A}}$ and $X_i^{\mathcal{A}}$, we add a function on these two variables to ensure that the arriving state of $X_{i-1}^{\mathcal{A}}$ is the starting state of $X_i^{\mathcal{A}}$:

$$c_{X_{i-1}^{\mathcal{A}}, X_i^{\mathcal{A}}} : ((q_{i-1}, q'_{i-1}), (q_i, q'_i)) \mapsto \begin{cases} 0 & \text{if } q'_{i-1} = q_i \\ \top & \text{otherwise.} \end{cases}$$

In the worst case, this function has size $O(\#D^2 \cdot \delta^2)$ ($O(\delta^2)$ in the Hamming case). Only $n$ extra variables are required.

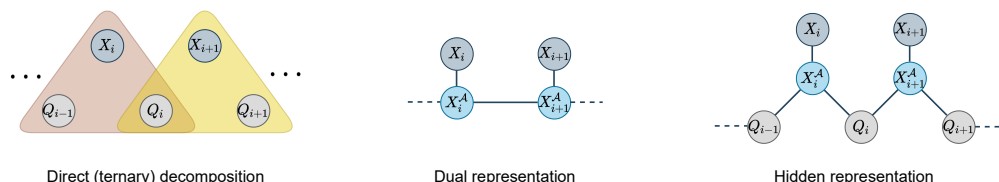

**Figure 5.** Representation of the ternary decomposition, and its dual and hidden representations.

The *hidden* representation of a constraint network $\mathcal{X} = (\mathbf{X}, \mathbf{D}, \mathbf{C})$ is a network $\mathcal{X}'' = (\mathbf{X}'', \mathbf{D}'', \mathbf{C}'')$ with:

- All the variables in $\mathbf{X}$ and the variables $X_{\mathbf{S}}$ from the dual network (and associated domains):

$$\mathbf{X}'' = \mathbf{X} \cup \mathbf{X}'$$

- For any dual variable $X_{\mathbf{S}}$, and each $X_i \in \mathbf{S}$, the set of constraints $\mathbf{C}''$ contains a function involving $X_i$ and $X_{\mathbf{S}}$:

$$c_{X_i X_{\mathbf{S}}} : (v, \mathbf{t}) \in \mathbf{D}_i \times \mathbf{D}_{X_{\mathbf{S}}} \mapsto \begin{cases} 0 & \text{if } \mathbf{t}[X_i] = v \\ \top & \text{otherwise.} \end{cases}$$

As before, this transformation is applied to the reduced $w^{\mathcal{A}}_{Q_i, X_{i+1}, Q_{i+1}}$ functions only (see Figure 5). In this new hidden representation, we keep variables $Q_i$ and create two pairwise functions involving each $Q_i$ and respectively $X_i^{\mathcal{A}}$ and $X_{i+1}^{\mathcal{A}}$:

$$c_{Q_i X_{i-1}^{\mathcal{A}}} : (q'', (q, q')) \mapsto \begin{cases} 0 & \text{if } q'' = q \\ \top & \text{otherwise.} \end{cases}$$

$$c_{Q_i X_i^{\mathcal{A}}} : (q'', (q, q')) \mapsto \begin{cases} 0 & \text{if } q'' = q' \\ \top & \text{otherwise.} \end{cases}$$

These functions ensure that the state value of $Q_i$ is consistent with the arriving state of the transition represented in $X_{i-1}^{\mathcal{A}}$ and the starting state of $X_i^{\mathcal{A}}$. In the worst case, these functions have size $O(\#D \cdot \delta^2)$ ($O(\delta^2)$ in the Hamming case).

The dedicated dual and hidden representations require the description of $O(\delta \cdot d + \#D^2 \cdot \delta^2)$ and $O(\delta \cdot d + \#D \cdot \delta^2)$ tuples respectively (it is $O(\delta \cdot d + \delta^2)$ in the Hamming case), instead of the $O(d \cdot \delta^2)$ tuples in $w^{\mathcal{A}}_{Q_i, X_{i+1}, Q_{i+1}}$.

## 7. Greedy DiverseSeq

The task at hand is the resolution of DIVERSESET$(\mathcal{C}, D, M, \delta)$, i.e., the generation of a set of $M$ solutions with minimum cost that satisfies a minimum pairwise diversity constraint. The exact computation being too expensive, we are tackling a greedy computation of DIVERSESEQ$(\mathcal{C}, D, M, \delta)$, a set of diverse good solutions that approximates DIVERSESET. The DIVERSESEQ computation is iterative:

1. The CFN $\mathcal{C}$ is solved using branch-and-bound while maintaining soft local consistencies [26].
2. If a solution $\mathbf{t}$ is found, it is added to the ongoing solution sequence $\mathbf{Z}$.
3. If $M$ solutions have been produced, the algorithm stops.
4. Otherwise, the cost function DIST$(\mathbf{X}, \mathbf{t}, D, \delta)$ is added to the previously solved problem.
5. We loop and solve the problem again (Step 1)

At Step 2, if no solution exists, the sequence of solutions $\mathbf{Z}$ can provably not be extended to length $M$ and the problem has no solution (but a shorter sequence has been produced).

This basic schema has been improved in three different ways (see Algorithm 1):

---

**Algorithm 1:** Incremental production of DiverseSeq$(\mathcal{C}, D, M, \delta)$

---

1  **Procedure** Solve $(\mathcal{C}, lb, ub)$
2  $\quad$ Compute optimum solution $\mathbf{t}^*$ of $\mathcal{C}$ with $lb \leqslant \mathsf{Cost}(\mathbf{t}^*) < ub$;
3  $\quad$ **if** $\mathbf{t}^*$ *exists* **then**
4  $\quad\quad$ $\mid$ **return** $(\mathbf{t}^*, true)$
5  $\quad$ **else**
6  $\quad\quad$ $\mid$ **return** $(\varnothing, false)$;

7  **Procedure** IncrementalSearch $(\mathcal{C}, lb, ub, \Delta^h, \mathbf{Z}, M, D, \delta)$
8  $\quad$ $\mathbf{t}^*, solved \leftarrow \mathsf{Solve}(\mathcal{C}, lb, \mathsf{Cost}(Z[i-1] + \Delta^h))$;
9  $\quad$ **if** *not solved* **then**
10 $\quad\quad$ $\mid$ $\mathbf{t}^*, solved \leftarrow \mathsf{Solve}(\mathcal{C}, lb, ub))$; $\qquad\qquad$ ▷ Upper bound prediction failed
11 $\quad$ **if** *not solved* **then**
12 $\quad\quad$ $\mid$ **return** $\mathbf{Z}$ ; $\qquad\qquad\qquad\qquad$ ▷ Z cannot be extended to length $M$
13 $\quad$ $\mathbf{Z} \leftarrow \mathbf{Z} \cup \{t^*\}$;
14 $\quad$ **if** $|\mathbf{Z}| = M$ **then**
15 $\quad\quad$ $\mid$ **return** $\mathbf{Z}$ ; $\qquad\qquad\qquad\qquad$ ▷ Enough solutions have been produced
16 $\quad$ Add $\mathsf{Dist}(\mathbf{X}, \mathbf{t}^*, D, \delta)$ to $\mathcal{C}$;
17 $\quad$ Propagate and save local consistencies in $\mathcal{C}$;
18 $\quad$ Update $\Delta^h$ ; $\qquad\qquad\qquad\qquad\qquad$ ▷ Using Cost $(\mathbf{t}^*)$
19 $\quad$ $lb \leftarrow \mathsf{Cost}(\mathbf{t}^*)$;
20 $\quad$ DiverseSeq $(\mathcal{C}, lb, ub, \Delta^h, \mathbf{Z}, M, D, \delta)$;

21 **Procedure** DiverseSeq $(\mathcal{C}, M, D, \delta)$
22 $\quad$ $\mid$ **return** IncrementalSearch$(\mathcal{C}, 0, \top, 0, \varnothing, M, D, \delta)$;

---

**Incrementality** Since the problems solved are increasingly constrained, all the equivalence preserving transformations and pruning that have been applied to enforce local consistencies at iteration $i - 1$ are still valid in the following iterations. Instead of restarting from a problem $\mathcal{C} = (\mathbf{X}, \mathbf{D}, \mathbf{C} \cup \bigcup_{1 \leq j < i} \{\mathsf{Dist}(\mathbf{X}, \mathbf{Z}[j], D, \delta)\})$, we reuse the problem solved at iteration $i - 1$ after it has been made locally consistent, add the $\mathsf{Dist}(\mathbf{X}, \mathbf{Z}[i-1], D, \delta)$ constraint and reinforce local consistencies. As with incremental SAT solvers, adaptive variable ordering heuristics that have been trained at iteration $i - 1$ are reused at iteration $i$.

**Lower bound** Since the problems solved are increasingly constrained, we know that the optimal cost $oc^i$ obtained at iteration $i$ cannot have a lower cost than the optimum cost $oc^{i-1}$ reported at iteration $i - 1$. When large plateaus are present in the energy landscape, this allows stopping the search as soon as a solution of cost $oc^{i-1}$ is reached, avoiding a useless repeated proof of optimality.

**Upper bound prediction** Even if there are no plateaus in the energy landscape, there may be large regions with similar variations in energy. In this case, the difference in energy between $oc^{i-1}$ and $oc^i$ will remain similar for several iterations. Let $\Delta_i^h = \max_{\max(2, i-h) \leq j < i}(oc^j - oc^{j-1})$ be the maximum variation observed in the last $h$ iterations (we used $h = 5$). At iteration $i$, we can first solve the problem with a temporary upper bound $k' = \min(k, oc_{i-1} + 2.\Delta_i^h)$ that should preserve a solution. If $k' < k$, this will lead to increased determinism, additional pruning, and possibly exponential savings. Otherwise, if no solution is found, the problem is solved again with the original upper bound $k$. We call this *predictive bounding*.

Each of these three improvements has the capacity to offer exponential time savings, and all are used in the experiments presented in the next sections.

## 8. Results

We implemented the iterative approach described above in its direct (ternary) decomposition, hidden and dual representations in the CFN open-source solver toulbar2 and experimented with it on various CFNs representing real Bayesian Networks [44]. All three decompositions offered comparable efficiency but empirically, as expected, the dual encoding was almost systematically more efficient. It is now the default for diversity encoding in toulbar2. All toulbar2 preprocessing algorithms dedicated to exact optimization that do not preserve suboptimal solutions were deactivated at the root node (variable elimination, dead-end elimination, variable merging). We chose to enforce strong virtual arc consistency (flag `-A` in toulbar2). The computational cost of VAC, although polynomial, is high, but amortized over the $M$ resolutions. During tree search, the default existential directional arc consistency (EDAC) was used. All experiments were performed on one core of a Xeon Gold 6140 CPU at 2.30 GHz. Wall-clock times could be further reduced using a parallel implementation of the underlying Hybrid Best-First search engine [45], currently under development in `toulbar2`.

Following our main motivation for protein design, we extracted two sets of prepared protein backbones for full redesigns from the benchmark sets built by [46,47] with the aim of checking if, as expected, diverse libraries can improve the overall design process. In the benchmark of monomeric proteins of less than 100 residues, with an X-ray resolved structure below 2 Å, with no missing or nonstandard residues and no ligand from [46], we selected the 20 proteins that had required the least CPU-time to solve, as indicated in the Excel sheet provided in the supplementary information of paper [46]. The harder instances from [47] correspond to proteins with diverse secondary structure compositions and fold classes. We selected the 17 instances that required less than 24 h of CPU-time for the full redesigned GMEC to be computed by toulbar2. These instances are listed in Table 1.

**Table 1.** List of protein structures used in our benchmark set, for full redesign: pdb identifier, domain length $n$ (number of variables in the resulting CFN) and maximum domain size $d$.

| PDB ID | $n$ | $d$ | PDB ID | $n$ | $d$ | PDB ID | $n$ | $d$ |
|--------|-----|-----|--------|-----|-----|--------|-----|-----|
| 1aho | 56 | 378 | 3i8z | 50 | 354 | 1ten | 81 | 392 |
| 2fjz | 53 | 324 | 2cg7 | 82 | 380 | 1ucs | 60 | 342 |
| 1b9w | 78 | 386 | 3rdy | 65 | 396 | 2bwf | 69 | 347 |
| 2gkt | 45 | 357 | 2erw | 47 | 446 | 2evb | 68 | 323 |
| 1f94 | 53 | 386 | 3vdj | 67 | 391 | 2o37 | 60 | 386 |
| 2pne | 77 | 401 | 2fht | 64 | 346 | 2o9s | 48 | 327 |
| 1hyp | 66 | 385 | 1bxy | 52 | 384 | 3f04 | 87 | 356 |
| 2pst | 61 | 357 | 1ctf | 68 | 349 | 3fym | 70 | 348 |
| 1uln | 66 | 367 | 1czp | 76 | 373 | 3gqs | 67 | 344 |
| 1uoy | 56 | 337 | 1fqt | 85 | 377 | 3gva | 87 | 348 |
| 2ca7 | 44 | 348 | 1guu | 47 | 350 | 3i2z | 67 | 360 |
| 1yzm | 46 | 294 | 1t8k | 68 | 361 | | | |

Full redesign was performed on each protein structure, and CFN instances were generated using the Dunbrack library [16] and Rosetta `ref2015` score function [48]. Alternate rotamer libraries and score functions can be used if required as the algorithms presented here are not specialized for Rosetta (and not even for CPD, see [44]). The resulting networks have from 44 to 87 rotamer variables, and maximum domain sizes range from 294 to 446 rotamers. The number of variables is doubled after sequence variables are added.

*Predictive bounding contribution*

$M = 10$ solutions with diversity threshold $\delta = 10$ for each problem from [46] were generated, with and without predictive bounding. The worst CPU-time spent on the resolution without predictive bounding was 32 min. It was reduced to 17 min with predictive bounding. The average computation time was 201s per problem. This shows that predictive bounding provides a simple and efficient boost and that real CPD instances

can be solved in a reasonable time, even when relatively large diversity requirements are used.

*Diversity improves prediction quality*

For all instances, sets of $M = 10$ solutions were generated with diversity threshold $\delta$ ranging from 1 to 15. For $\delta = 1$, the set of solutions produced is just the set of the 10 best (minimum energy) sequences.

These CPD problems use real protein backbones, determined experimentally. A native sequence exists for these backbones, therefore it is possible to measure the improvements diversity brings in terms of recovering native sequences, known to be folded and functional. Two measures are often used to assess computational protein design methods. The Native Sequence Recovery (NSR) is the percentage of amino acid residues in the designed protein which are identical to the amino acid residues in the native sequence that folds on the target backbone. The NSR can be enhanced by taking into account *similarity scores* between the amino acid types. Such scores are provided by similarity matrices, such as BLOSUM62 [30]. The *Native Sequence Similarity Recovery* (NSSR) is the fraction of positions where the designed and native sequences have a positive similarity score. NSR and NSSR measure how much the redesigned protein resembles the natural in terms of sequence. Although often used, these measures have their own limitations: while protein design targets maximal stability, natural protein only require sufficient marginal stability. In the end, they therefore provide a useful but imperfect proxy for computation protein design evaluation: a perfect (100%) recovery would not necessarily indicate the best algorithm (also because the approximate energy function plays a major role here).

If solution diversity helps, the maximum NSR/NSSR over the 10 sequences should improve when $\delta$ is large compared to when $\delta = 1$, as long as the costs remain close to the optimum. A solution cost too far from the optimum, which could be generated because of a diversity threshold set too high, would mean a poor quality of the solution. Even with $\delta = 15$, the maximum difference in energy we observed with the global minimum energy never exceeded 4.3 kcal/mol (with an average of 2.1 kcal/mol).

For each protein, and each diversity threshold $\delta = 2\ldots15$, we compared the best NSR (resp. NSSR) that could be obtained with $\delta$ to the best obtained with diversity threshold 1, i.e., the simple enumeration of the 10 best sequences. Results are plotted in Figure 6 (resp. Figure 7). Although somewhat noisy, they show a clear general increase in the best NSR (resp. NSSR) when diverse sequences are produced, even with small $\delta$. To validate the statistical significance of the diversity improvement of sequence quality, *p*-values for a unilateral Wilcoxon signed-rank test comparing the sample of best NSR (resp. NSSR) for each $\delta = 2\ldots15$ with $\delta = 1$ were computed. They are shown in Table 2 and confirm the improvement brought by increasingly diverse libraries.

The improvements are more clearly visible when one compares the absolute improvement in NSR (and NSSR) obtained when one compares the best sequences produced inside a library using a guaranteed diversity of 15 versus just 1, as illustrated in Figure 8. On most backbones (X-axis), the increased diversity yields a clear improvement in the NSR (Y-axis), with the largest absolute improvement exceeding 15% in NSR. For a small fraction of backbones, there is absolutely no improvement. These are backbones for which the GMEC actually provides the best NSR in both the 1-diverse and the 15-diverse cases. Then an even smaller fraction of backbones shows an actual decrease in NSR: one close to GMEC solution did better than any of the 15-diverse sequences. The degradation here is very limited and likely corresponds to substitutions to similar amino acids. This is confirmed by the NSSR curve that takes into account amino acid properties through the underlying BLOSUM62 matrix used. Here, only one case shows a degradation in NSSR.

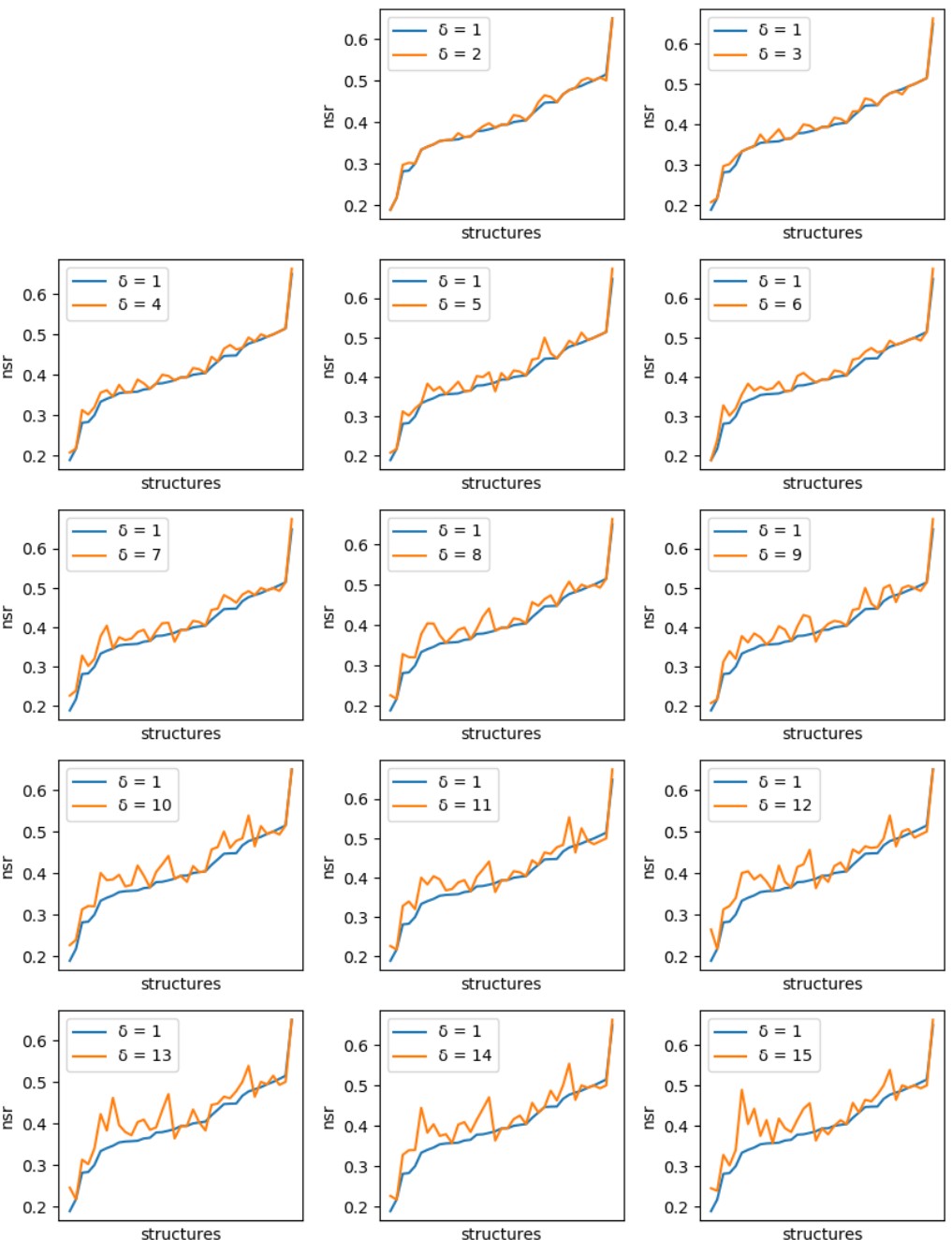

**Figure 6.** Comparison of the best NSR value obtained with ten 1-diverse sequences ($\delta = 1$, blue curve) with the best NSR value obtained with libraries of ten sequences of increased diversity. Each plot corresponds to a specific additional value of $\delta$ ($\delta = 2$ to 15, golden curve). Plots are ordered lexicographically from top-left to bottom-right, with increasing values of diversity ($\delta$). In each plot, the X-axis ranges over all tested backbones, sorted in increasing order of NSR value for the 1-diverse case and the Y-axis gives the corresponding NSR value. As the diversity requirement increases, the NSR value indicated by the golden curve increases also visibly.

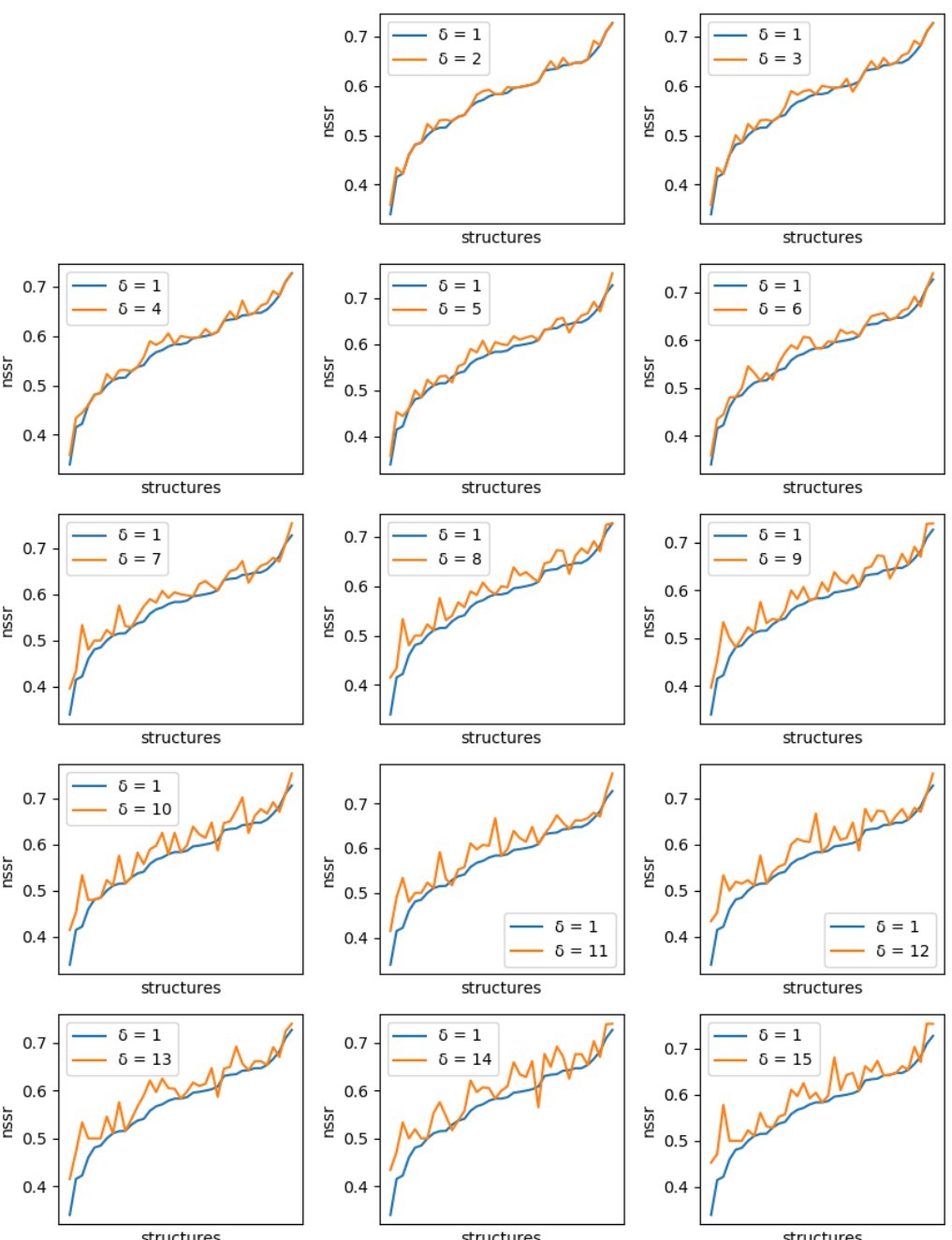

**Figure 7.** Comparison of the best NSSR value obtained with ten 1-diverse sequences ($\delta = 1$, blue curve) with the best NSSR value obtained with libraries of ten sequences of increased diversity. Each plot corresponds to a specific additional value of $\delta$ ($\delta = 2$ to 15, golden curve). Plots are ordered lexicographically from top-left to bottom-right, with increasing values of diversity ($\delta$). In each plot, the X-axis ranges over all tested backbones, sorted in increasing order of NSSR value for the 1-diverse case and the Y-axis gives the corresponding NSSR value. As the diversity requirement $\delta$ increases, the NSSR value indicated by the golden curve increases also visibly.

**Table 2.** *p*-values for a unilateral Wilcoxon signed-rank test comparing the sample of best NSR (resp. NSSR) for each $\delta = 2 \ldots 15$ with $\delta = 1$, for optimal and suboptimal (3 kcal/mol allowed energy gap to real optimum) resolution.

| $\delta$ | Exact Resolution | | Subopt. Resolution | |
|---|---|---|---|---|
| | **NSR** | **NSSR** | **NSR** | **NSSR** |
| 2 | $2.88 \times 10^{-3}$ | $1.10 \times 10^{-3}$ | $4.11 \times 10^{-1}$ | $1.35 \times 10^{-1}$ |
| 3 | $3.87 \times 10^{-4}$ | $1.01 \times 10^{-4}$ | $1.14 \times 10^{-1}$ | $2.60 \times 10^{-2}$ |
| 4 | $4.42 \times 10^{-5}$ | $6.58 \times 10^{-5}$ | $4.48 \times 10^{-3}$ | $1.06 \times 10^{-3}$ |
| 5 | $8.11 \times 10^{-5}$ | $1.54 \times 10^{-5}$ | $1.98 \times 10^{-3}$ | $2.15 \times 10^{-3}$ |
| 6 | $1.51 \times 10^{-5}$ | $4.39 \times 10^{-6}$ | $7.47 \times 10^{-5}$ | $4.49 \times 10^{-5}$ |
| 7 | $1.88 \times 10^{-5}$ | $4.23 \times 10^{-6}$ | $8.86 \times 10^{-6}$ | $3.50 \times 10^{-5}$ |
| 8 | $1.27 \times 10^{-5}$ | $1.49 \times 10^{-6}$ | $8.19 \times 10^{-6}$ | $1.77 \times 10^{-5}$ |
| 9 | $2.76 \times 10^{-5}$ | $5.97 \times 10^{-6}$ | $2.07 \times 10^{-5}$ | $2.80 \times 10^{-6}$ |
| 10 | $1.14 \times 10^{-5}$ | $1.18 \times 10^{-5}$ | $1.78 \times 10^{-5}$ | $3.06 \times 10^{-5}$ |
| 11 | $4.27 \times 10^{-5}$ | $5.81 \times 10^{-7}$ | $2.32 \times 10^{-5}$ | $1.73 \times 10^{-5}$ |
| 12 | $6.63 \times 10^{-5}$ | $2.26 \times 10^{-6}$ | $1.75 \times 10^{-5}$ | $1.18 \times 10^{-5}$ |
| 13 | $4.43 \times 10^{-5}$ | $2.52 \times 10^{-6}$ | $2.48 \times 10^{-6}$ | $6.15 \times 10^{-6}$ |
| 14 | $2.29 \times 10^{-5}$ | $5.76 \times 10^{-6}$ | $5.26 \times 10^{-6}$ | $3.89 \times 10^{-7}$ |
| 15 | $3.92 \times 10^{-5}$ | $1.58 \times 10^{-6}$ | $2.68 \times 10^{-5}$ | $4.86 \times 10^{-5}$ |

Diversity also has the advantage that it is more likely to provide improved sequences when the predicted 1-diverse sequences are poor. Indeed, with an initial sequence with NSR equal to $r$, the introduction of a random mutation will move us away from the native in $r$% of cases (we mutate a correct amino acid) and otherwise (we mutate a wrong amino acid) leave us with a wrong amino acid again (in 18/19 cases, leaving the NSR unchanged) or bring us closer to the native sequence with 1/19 probability. On average, a mutation should therefore decrease the number of correct positions with probability $(r - \frac{1-r}{19})$, which increases with $r$ and is positive as soon as the sequence has NSR higher than 5% ($\frac{1}{20}$). Our results confirm this trend, as shown in the NSR figure on the left of Figure 8: among the ten backbones with the highest improvement in NSR, nine had a 1-diverse NSR below the average 1-diverse NSR. Conversely, only 50% of the ten less improved backbones had a below-average 1-diverse NSR. Although these improvements underline the approximate nature of the energy function, showing that it is often worth to explore the sequence space beyond the GMEC, they also confirm that energy, on average, does guide us towards native sequences: instead of degrading NSR as soon as $r > \frac{1}{20}$, energy optimization pushes the introduced mutations to improve NSR in most cases, even with 15 introduced mutations and initial NSRs ranging from 20 to 60%.

Each dissimilarity constraint adds $n$ extra variables to the network (with the dual representation). These variable domain sizes increase with the diversity threshold $\delta$ and contribute to the construction of increasingly large CFN instances that need to be solved. Computation times are plotted in Figure 9. As expected, a threshold increase leads to an exponential increase in the computation time. The small set of points on the top right corner of the plot correspond to the protein structure 3FYM. This protein, with 70 amino acids, is not the largest in our benchmark set, a reminder that size is not necessarily the best predictor of empirical computational cost in NP-hard problem solving. On this backbone, for high diversity thresholds, the 24 h computation time limit was reached and less than 10 sequences were produced.

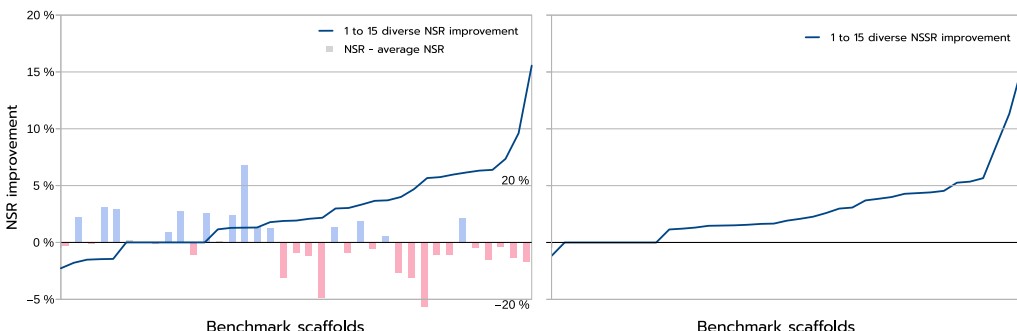

**Figure 8.** The blue curves above give the absolute change in NSR (Y-axis, **left** figure) and NSSR (Y-axis, **right** figure) between the best 15-diverse and the best 1-diverse sequences found for each backbone. Backbones (on the X-axis) are ordered in increasing order of the corresponding measure. In the left figure, the bar-plot shows the difference between each backbone 1-diverse NSR and average 1-diverse NSR over all backbones. The corresponding NSR change scale appears on the right with $\pm 20\%$ labels. Red bars indicate a below-average 1-diverse NSR while blue bars indicate an above average 1-diverse NSR. The most improved NSRs, on the right of the left figure, mostly appear on weak (red, below average) 1-diverse NSRs.

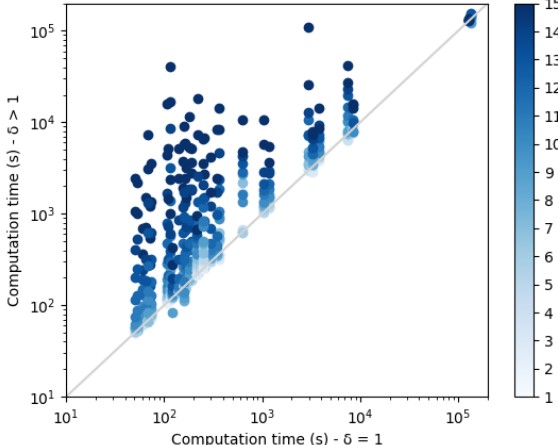

**Figure 9.** Comparison of the computation times of sequence sets without diversity $\delta = 1$, with sequence sets with diversity $\delta > 1$. The color scale on the right indicates the corresponding value of $\delta$.

Given that the optimized function is an approximation of the real intractable energy, solving the problem to optimality might seem exaggerated. The requirement for optimality that we have used in previous experiments can be trivially relaxed to a relative or absolute approximation guarantee using artificially tightened pruning rules as originally proposed in [49] in the Weighted-A* algorithm. This pruning mechanism is already implemented in the toulbar2 solver (using the `-rgap` and `-agap` flags respectively).

For diversity threshold $\delta = 2 \ldots 15$, we generated sets of 10 suboptimal diverse sequences that satisfy the diversity constraints, but allowing for a 3 kcal/mol energy gap to the optimum. Our optimizations are still provable, but the optimality guarantee is now reduced to a bounded sub-optimality guarantee. Empirically, the maximum energy degradation we observed with the global minimum energy over the 10 diverse sequences produced never exceeded 5.65 kcal/mol (with an average energy difference of 3.86 kcal/mol). This is only slightly more than the 4.3 kcal/mol worse degradation (average 2.1 kcal/mol) of the resolution, when exact optimum are used.

We compared these samples of suboptimal sequences to the set of 10 exact best sequences. Results for NSR and NSSR are shown in Figures 10 and 11 respectively, and corresponding *p*-values are displayed in Table 2 (unilateral Wilcoxon signed-rank test).

With dissimilarity threshold $\delta \geqslant 6$, it is clear that the set of diverse suboptimal sequences have better quality than the 10 best enumerated sequences. Moreover, as shown in Figure 12, for harder instances, suboptimal diverse resolution becomes faster than exact enumeration.

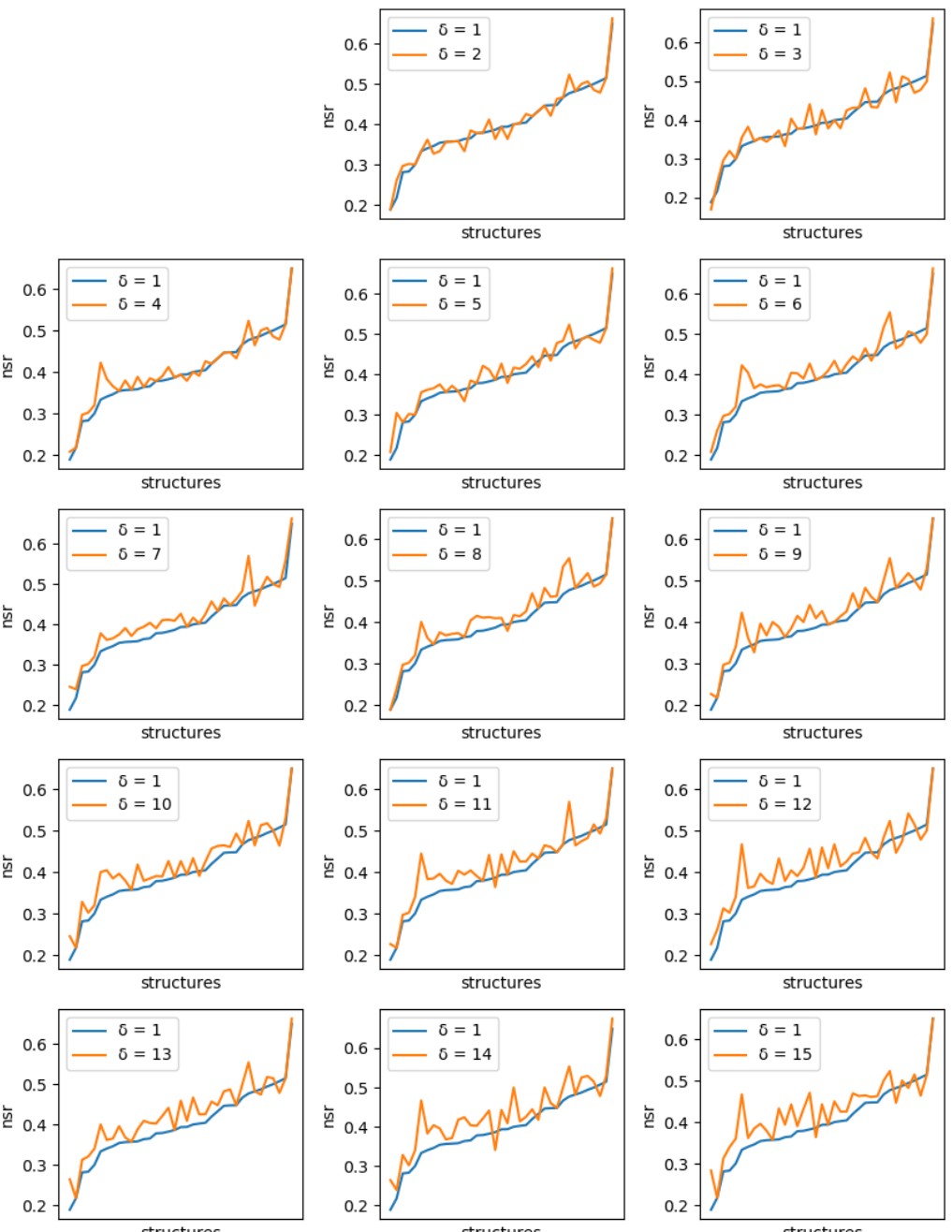

**Figure 10.** Comparison of the best NSR value obtained with ten 1-diverse sequences ($\delta = 1$, blue curve) with the best NSR value obtained with libraries of ten sequences of increased diversity all predicted with an allowed gap top optimal energy of 3 kcal/mol. Each plot corresponds to a specific additional value of $\delta$ ($\delta = 2$ to 15, golden curve). Plots are lexicographically ordered from top-left to bottom-right, with increasing values of diversity ($\delta$). In each plot, the X-axis ranges over all tested backbones, sorted in increasing order of NSR value for the 1-diverse case and the Y-axis gives the corresponding NSR value. As the diversity requirement $\delta$ increases, the NSR value indicated by the golden curve increases also visibly.

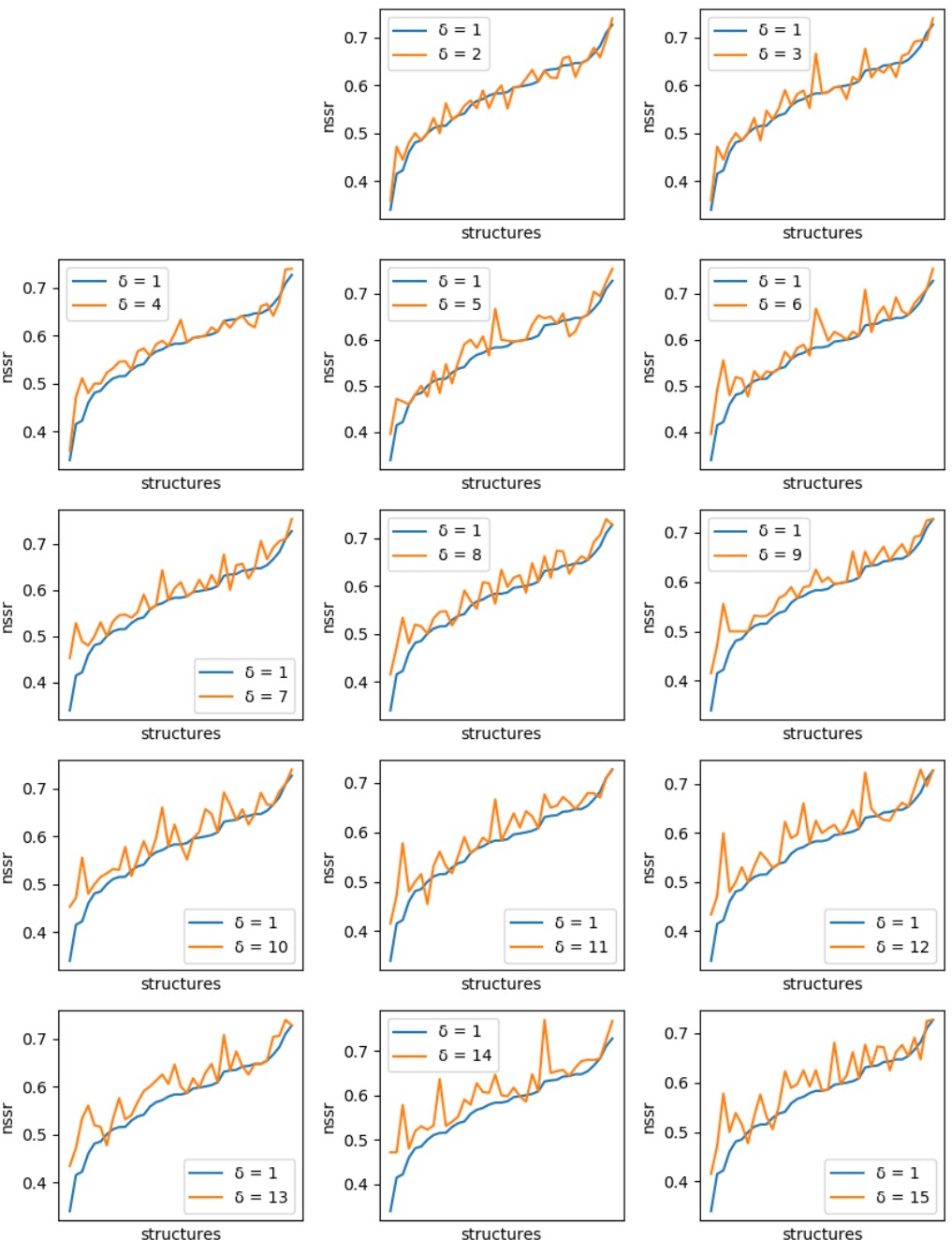

**Figure 11.** Comparison of the best NSSR value obtained with ten 1-diverse sequences ($\delta = 1$, blue curve) with the best NSSR value obtained with libraries of ten sequences of increased diversity all predicted with an allowed gap top optimal energy of 3 kcal/mol. Each plot corresponds to a specific additional value of $\delta$ ($\delta = 2$ to 15, golden curve). Plots are lexicographically ordered from top-left to bottom-right, with increasing values of diversity ($\delta$). In each plot, the X-axis ranges over all tested backbones, sorted in increasing order of NSSR value for the 1-diverse case and the Y-axis gives the corresponding NSSR value. As the diversity requirement $\delta$ increases, the NSSR value indicated by the golden curve increases also visibly.

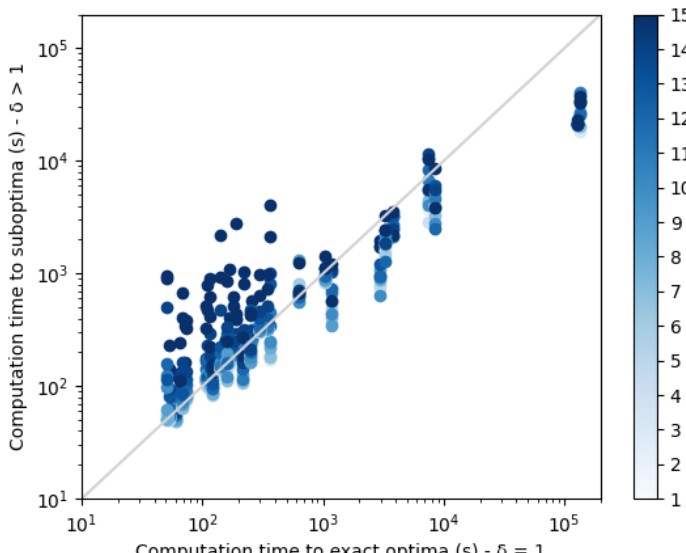

**Figure 12.** Comparison of the computation times of sequence sets without diversity $\delta = 1$, with suboptimal sequence sets with diversity $\delta > 1$. An energy gap of 3 kcal/mol is allowed for actual optimum.

Therefore, when predicting a library of sequences, if the instance is hard, it seems empirically wise to generate suboptimal diverse sequences, instead of enumerating minimum energy sequences, without diversity. Doing so, there is a higher chance of predicting better sequences "in practice" faster.

## 9. Conclusions

Producing a library of diverse solutions is a very usual requirement when an approximate or learned model is used for optimal decoding. In this paper, we show that with an incremental provable CFN approach that directly tackles a series of decision NP-complete problems, using diversity constraints represented as weighted automata that are densely encoded in a dedicated dual encoding together with predictive bounding, it is possible to produce sequences of solutions that satisfy guarantees on diversity on realistic full redesign computational protein design instances. This guarantee is obtained on dense problems, with non-permutated-submodular functions while also guaranteeing that each new solution produced is the best given the previously identified solutions.

We also showed that the stream of diverse solutions that our algorithm produces can be filtered and each solution efficiently identified as being a $\delta$-mode or not. $\delta$-mode represent local minima, each defining its own basin in the protein sequence energy landscape. Beyond their direct use for design, the guaranteed diversity provided by our algorithm could also be of interest to perform more systematic analyses of such energy landscapes.

On real protein design problems, we observe that small and large diversity requirements do improve the quality of sequence libraries when native proteins are fully redesigned. Moreover, large diversity requirements on suboptimal sequences also improve the quality of sequence libraries, compared to a simple enumeration of the minimum energy sequences. In the context of optimizing an approximate or learned function, the requirement for an optimal cost solution may be considered to be exaggerated. Our guaranteed suboptimal resolution is useful, given that even computationally expensive approaches with asymptotic convergence results such as simulated annealing may fail with unbounded error [46].

Two directions could extend this work. Beyond the language of permutated submodular functions, the other important tractable class of CFN is the class of CFN with a graph of bounded tree-width. This parameter is exploited in several protein packing [50] and

design [51] algorithms and is also exploited in dedicated branch-and-bound algorithms, also implemented in toulbar2 [45,52]. These tree-search algorithms can trade space for time and are empirically usable on problems with a tree-width that is too large to make pure dynamic programming applicable, mostly because of its space-complexity (in $O(d^w)$ where $d$ is the domain size and $w$ is the width of the tree-decomposition used). On such problems, it would be desirable to show that the decomposed ternary or binary functions we use for encoding DIST can be arranged in such a way that tree-width can be preserved or more likely, not exaggeratedly increased. This would enable the efficient production of diverse solutions for otherwise unsolved structured instances.

Another direction would be to identify a formulation of the DIST (and possibly $\text{DIV}_{min}$) constraints that would provide better pruning or avoid the introduction of extra variables that often disturb dynamic variable ordering heuristics. One possibility would be to encode these using linear (knapsack) constraints for which dedicated propagators would need to be developed.

**Author Contributions:** Conceptualization, S.B. and T.S.; methodology, M.R., G.K. and T.S.; software, M.R., J.V.; S.d.G. and T.S.; validation, M.R.; formal analysis, M.R.; resources, T.S. and S.B.; data curation, M.R., J.V., T.S. and S.B.; writing—original draft preparation, M.R.; writing—review and editing, M.R., T.S. and S.B.; supervision, T.S. and S.B.; funding acquisition, T.S. All authors have read and agreed to the published version of the manuscript.

**Funding:** This research was funded by the French "Agence Nationale de la Recherche" through grants ANR-16-CE40-0028 and ANR-19-PI3A-0004.

**Institutional Review Board Statement:** Not applicable.

**Informed Consent Statement:** Not applicable.

**Data Availability Statement:** No data specific to this paper. The benchmarks we use are available on existing repositories and the code of toulbar2, which includes our contributions, is available on GitHub at https://github.com/toulbar2/toulbar2 (from version 1.1.1, accessed on 20 November 2020) under an OSI MIT license.

**Acknowledgments:** We thank the GenoToul (Toulouse, France) Bioinformatics platform and the CALMIP HPC platform for their computational support.

**Conflicts of Interest:** The authors declare no conflict of interest.

## Abbreviations

The following abbreviations are used in this manuscript:

| | |
|---|---|
| CFN | Cost Function Network |
| CPD | Computational Protein Design |
| CSP | Constraint Satisfaction Problem |
| MRF | Markov Random Field |
| NSR | Native Sequence Recovery |
| NSSR | Native Sequence Similarity Recovery |
| WCSP | Weighted Constraint Satisfaction Problem |

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
