# Peer review of "Guaranteed Diversity and Optimality in Cost Function Network Based Computational Protein Design Methods"

_algorithms, doi:10.3390/a14060168_

Round 1

Reviewer 1 Report

In “Guaranteed Diversity and Optimality in Cost Function Network Based Computational Protein Design Methods”, Ruffini and coworkers introduce a new method for solving the optimization problem that one solves when designing a protein. Where most existing methods seek only a single solution for the amino acid sequence that yields the lowest value of some objective function (typically an energy function), the method of Ruffini et al. provides users with a means of generating a pool of solutions with strong guarantees of both optimality and diversity. This method is benchmarked on a set of protein design problems, using native sequence recovery (NSR) and native sequence similarity recovery (NSSR) scores to evaluate it. Importantly, the authors have made their algorithm publicly available in the Toulbar2 software, with a permissive licence allowing others to use, evaluate, or build on this method.

I believe that anyone who has spent any time designing proteins will see the considerable benefit of this new approach. Protein designers have at their disposal wet-lab screening techniques that allow hundreds or thousands of designs to be evaluated. While this is a small number compared to the size of the solution space for the sequence design problem, it is large enough that a computational designer does seek to produce a pool of designs, not just a single design, to maximize probability of success (particularly given that the objective function optimized during design is approximate and imperfect, and the best design in reality may only be a near-optimal design as scored by the objective function). Traditional design approaches that use stochastic methods such as Monte Carlo-based searches of rotamer space often either converge to the same solution repeatedly (for small problems), or yield a diverse pool of sub-optimal solutions with no means of being confident about how close to optimal the solutions are. A method with strong guarantees of both optimality and diversity fills a current need central to the protein design field. The manuscript is very well written, clearly explaining the mathematical reasoning in a manner that biologists and biochemists can understand, as well as introducing the salient concepts in protein design in a manner suitable for mathematicians and computer scientists. Overall, I think this work makes a significant contribution to the protein design field. I believe that it will be suitable for publication pending minor revisions which I think the authors will be able to make easily. My suggestions are listed below.

Larger issues:

1. In the introduction, the authors correctly point out that one of the reasons for desiring diversity is that the energy function that is typically optimized is approximate (“the optimized pairwise decomposed energetic criterion only approximates the actual molecule energy”). There is a second reason to desire diversity that should probably also be mentioned: even given a perfect energy function, the sequence that minimizes energy in the desired conformation is not necessarily the sequence that ensures that the desired conformation is the global minimum energy conformation. Indeed, this is a frequent problem in the protein design field: often the sequence yielding the lowest energy also stabilizes many alternative backbone conformations of the protein. Ideally, one wants the sequence yielding the largest energy gap between the desired backbone conformation and all alternative backbone conformations. Since it is computationally intractable to consider all alternative backbone conformations during design, the approach used – seeking the sequence that yields the lowest energy in the desired backbone conformation – is substituted as an approximation. Often, one performs a subsequent backbone conformation sampling step on one’s best designs (keeping sequences fixed) for validation, to evaluate which ones have large energy gaps, and this is another reason to desire diversity in one’s pool of sequences. I think that it is worth mentioning this point (particularly since it adds to the significance of the algorithm that the authors introduce here).

2. The statement that “The resulting sequence-conformation [from fixed-backbone sequence design] is called the Global Minimum Energy Conformation” should probably be reworded or omitted, since it is not true that the sequence that the sequence yielding the lowest energy in the desired backbone conformation necessarily ensures that that backbone conformation is a minimum-energy backbone conformation. (Alternatively, perhaps the confusion is in the word “conformation”, which can mean “side-chain conformation” or “backbone conformation”. If the former is meant, this should be clarified.)

3. Native sequence recovery (or the related native sequence similarity recovery) are not necessarily ideal metrics for assessing the quality of a sequence optimization algorithm. This is because native sequences are optimized by evolution for marginal stability (since proteins must be degraded at the end of their life cycle) and not for maximal stability, meaning that the hypothetical “best” energy function and “best” optimization algorithm could conceivably find sequences that are far more stable than those of natural proteins, possessing little sequence identity or similarity. (In practice, protein designers now routinely design and produce proteins that are far more stable than natural proteins.) The limits of NSR and NSSR as metrics are a problem for the protein design field as a whole, since it’s hard to devise good metrics to use. While I think it is appropriate for the authors to use NSR and NSSR (particularly since these provide a point of comparison to other algorithms that were benchmarked in this way), it would be good to at least mention that (a) perfect reproduction of native sequences would not necessarily be a hallmark of the best algorithm, and (b) that these metrics evaluate both the optimization method and the energy function used (in this case, the Rosetta ref2015 energy function), and that the imperfections of the latter could hinder the performance of the former as measured by these metrics.

4. One of the strengths of this work is its generality: the approach is not dependent on, say, the Dunbrack libraries, or the Rosetta ref2015 energy function, or even the assumption that proteins must be built from the 20 canonical amino acid types. I think it would be a good idea to stress that point explicitly somewhere, possibly either in the introduction or the conclusion: although the test cases happen to use the Dunbrack libraries and Rosetta ref2015 energies and canonical amino acids, this is just as easily applicable to arbitrary heteropolymer design problems using any discretization of side-chain rotamers and any energy or scoring function (albeit with advantages for pairwise energy functions).

Minor suggestions:

1. In the introduction, it is asserted that “all amino acids share a common constant linear body and a variable side-chain”. Strictly, all α-amino acids (alpha-amino acids) share a common backbone. Natural proteins are made only of α-amino acids, but there do exist amino acids with longer or more exotic backbones that can be used in synthetic heteropolymer design. (Adding α ["alpha"] to the second and third sentences of the introduction should address this; a lengthy discussion of exotic amino acids would likely be beyond the scope of this work.)

2. On line 37 of the introduction, “atomic forces that control the protein stability are represented as a decomposable energy function, defined as the sum of terms involving at most two bodies (atoms)”. That should be “at most two bodies (amino acids)”: energy functions often are not atom-level pairwise-decomposable. (In the Rosetta energy function, for example, there are one- and two-body torsional terms that are dependent on four atoms, but only on one or two amino acids. Older Rosetta energy functions, such as score12, also included a “pair” potential dependent only on the identities of amino acids in close proximity. This potential would be dependent on dozens of atoms but on only two amino acids.)

3. On line 65 of the introduction, the term “energy gap” is a bit confusing, since this has other meanings in protein folding (usually, the difference in energy between a desired backbone conformation and all alternative conformations). I’d suggest “threshold”, the term used in the previous sentence.

4. It may be worth mentioning in the introduction that protein designers can now express and screen thousands of designs in yeast display libraries, as another reason that computer algorithms that produce diverse designs are more practical and useful than computer algorithms that produce only a single top-scoring design. Library design is a reason to use a diversity algorithm.

5. Patented sequences are given as one reason that a designer may wish to impose a requirement that designs have a minimum Hamming distance from a known sequence. Another reason that the authors may or may not wish to mention is antigenicity: there are known short peptide sequences (sub-sequences of a longer protein) that are recognized by the Major Histocompatibility Complexes (MHCs). If one wants to make a protein that could be used as a drug, it needs to evade the human immune system, which means that one wants sequences for which all sub-sequences have a Hamming distance greater than a given threshold from known antigenic sequences. Some computational methods exist for redesigning good designs to remove antigenic sub-sequences while trying to retain structure and function (e.g. King et al. (2014). Proc. Natl. Acad. Sci. USA 111(23):8577-8582), but means of preventing immunogenicity during design while preserving optimality guarantees would be very powerful. I leave it to the author’s discretion whether mentioning this would help the paper, or whether it would add unnecessary complexity.

6. Since the Rosetta ref2015 energy function is used, Rebecca Alford’s 2017 paper about this energy function (Alford et al. (2017) J. Chem. Theory Comput. 13(6):3031-3048) should probably be cited.

7. On line 130, it is asserted that the computation of pairwise interaction energies is quadratic in protein length. This is actually a worst-case scaling. Since only positions close in space are considered to interact, and since each core residue has on average a constant number of neighbours, the scaling in practice tends to be O(N) with length instead of O(N^2). (The scaling is O(N^2) with number of rotamers per position, however).

8. On line 183, I think “solutions” should be possessive: “while maintaining the solutions’ joint costs unchanged”.

9. In Definition 4, the authors state “Allowing both positive and negative threshold delta allows he DIST cost function to express either minimum or maximum diversity constraints.” It may be good to clarify whether it is possible to impose a minimum and a maximum for the same problem.

10. On line 282, the authors state, “We empirically confirmed this on very tiny instances.” It may be useful to state how tiny. At what point does the scaling make this approach impractical?

11. On line 387, should “Figure 5” be “Figure 4”? (“Additional variables and ternary functions are represented in Figure 5.”)

12. In the first paragraph of Section 8, it is stated that the tests were performed on a single core of a Xeon Gold 6140 CPU at 2.30 GHz. (The authors’ attention to detail is commendable.) Have the authors given any thought as to whether this algorithm can be efficiently parallelized? Implementing a parallel version is likely beyond the scope of this work, but a few words about the feasibility might be useful.

13. In Figures 9 and 12, the colour key should be labelled. (Diversity or delta, I believe, if I am interpreting these figures correctly.)

14. This is a stylistic point that I would also leave to the authors’ discretion: although it is intellectually honest and commendable of the authors to discuss current limitations and future directions in their concluding remarks, the last sentence of the paper ends on a bit of a pessimistic-sounding note: “...we have been unable to reach an empirically convincing result here,” (referring to a particular detail that might be explored further in the future, not to the overall work). It may be worthwhile to add a couple of additional sentences to end with the significance of the work in its present form. I do think this paper represents a significant contribution to protein design in its present form, and I wouldn’t want a reader to end with the wrongful impression that something more needs to be done in order for this work to be useful: it is unambiguously useful right now.

Author Response

Review 1

In “Guaranteed Diversity and Optimality in Cost Function Network Based Computational Protein Design Methods”, Ruffini and coworkers introduce a new method for solving the optimization problem that one solves when designing a protein. Where most existing methods seek only a single solution for the amino acid sequence that yields the lowest value of some objective function (typically an energy function), the method of Ruffini et al. provides users with a means of generating a pool of solutions with strong guarantees of both optimality and diversity. This method is benchmarked on a set of protein design problems, using native sequence recovery (NSR) and native sequence similarity recovery (NSSR) scores to evaluate it. Importantly, the authors have made their algorithm publicly available in the Toulbar2 software, with a permissive licence allowing others to use, evaluate, or build on this method.

I believe that anyone who has spent any time designing proteins will see the considerable benefit of this new approach. Protein designers have at their disposal wet-lab screening techniques that allow hundreds or thousands of designs to be evaluated. While this is a small number compared to the size of the solution space for the sequence design problem, it is large enough that a computational designer does seek to produce a pool of designs, not just a single design, to maximize probability of success (particularly given that the objective function optimized during design is approximate and imperfect, and the best design in reality may only be a near-optimal design as scored by the objective function). Traditional design approaches that use stochastic methods such as Monte Carlo-based searches of rotamer space often either converge to the same solution repeatedly (for small problems), or yield a diverse pool of sub-optimal solutions with no means of being confident about how close to optimal the solutions are. A method with strong guarantees of both optimality and diversity fills a current need central to the protein design field. The manuscript is very well written, clearly explaining the mathematical reasoning in a manner that biologists and biochemists can understand, as well as introducing the salient concepts in protein design in a manner suitable for mathematicians and computer scientists. Overall, I think this work makes a significant contribution to the protein design field. I believe that it will be suitable for publication pending minor revisions which I think the authors will be able to make easily. My suggestions are listed below.

Larger issues:

  1. In the introduction, the authors correctly point out that one of the reasons for desiring diversity is that the energy function that is typically optimized is approximate (“the optimized pairwise decomposed energetic criterion only approximates the actual molecule energy”). There is a second reason to desire diversity that should probably also be mentioned: even given a perfect energy function, the sequence that minimizes energy in the desired conformation is not necessarily the sequence that ensures that the desired conformation is the global minimum energy conformation. Indeed, this is a frequent problem in the protein design field: often the sequence yielding the lowest energy also stabilizes many alternative backbone conformations of the protein. Ideally, one wants the sequence yielding the largest energy gap between the desired backbone conformation and all alternative backbone conformations. Since it is computationally intractable to consider all alternative backbone conformations during design, the approach used – seeking the sequence that yields the lowest energy in the desired backbone conformation – is substituted as an approximation. Often, one performs a subsequent backbone conformation sampling step on one’s best designs (keeping sequences fixed) for validation, to evaluate which ones have large energy gaps, and this is another reason to desire diversity in one’s pool of sequences. I think that it is worth mentioning this point (particularly since it adds to the significance of the algorithm that the authors introduce here).

We absolutely agree and also use “forward folding” as a filtering procedure in most of our designs. These points have been added in the introduction and a reference to a paper describing (biased) “Forward folding” included too.

  1. The statement that “The resulting sequence-conformation [from fixed-backbone sequence design] is called the Global Minimum Energy Conformation” should probably be reworded or omitted, since it is not true that the sequence that the sequence yielding the lowest energy in the desired backbone conformation necessarily ensures that that backbone conformation is a minimum-energy backbone conformation. (Alternatively, perhaps the confusion is in the word “conformation”, which can mean “side-chain conformation” or “backbone conformation”. If the former is meant, this should be clarified.)

Agreed of course. The sentence has been suitably clarified, hopefully. 

  1. Native sequence recovery (or the related native sequence similarity recovery) are not necessarily ideal metrics for assessing the quality of a sequence optimization algorithm. This is because native sequences are optimized by evolution for marginal stability (since proteins must be degraded at the end of their life cycle) and not for maximal stability, meaning that the hypothetical “best” energy function and “best” optimization algorithm could conceivably find sequences that are far more stable than those of natural proteins, possessing little sequence identity or similarity. (In practice, protein designers now routinely design and produce proteins that are far more stable than natural proteins.) The limits of NSR and NSSR as metrics are a problem for the protein design field as a whole, since it’s hard to devise good metrics to use. While I think it is appropriate for the authors to use NSR and NSSR (particularly since these provide a point of comparison to other algorithms that were benchmarked in this way), it would be good to at least mention that (a) perfect reproduction of native sequences would not necessarily be a hallmark of the best algorithm, and (b) that these metrics evaluate both the optimization method and the energy function used (in this case, the Rosetta ref2015 energy function), and that the imperfections of the latter could hinder the performance of the former as measured by these metrics.

Agreed again. NSR and NSSR are convenient proxies and the NSR/NSSR definition section has been refined with additional contextual information.

  1. One of the strengths of this work is its generality: the approach is not dependent on, say, the Dunbrack libraries, or the Rosetta ref2015 energy function, or even the assumption that proteins must be built from the 20 canonical amino acid types. I think it would be a good idea to stress that point explicitly somewhere, possibly either in the introduction or the conclusion: although the test cases happen to use the Dunbrack libraries and Rosetta ref2015 energies and canonical amino acids, this is just as easily applicable to arbitrary heteropolymer design problems using any discretization of side-chain rotamers and any energy or scoring function (albeit with advantages for pairwise energy functions).

We have been beyond this and quickly reminded the reader that these encodings are not specific or tuned for Rosetta-based design nor to CPD actually.

Minor suggestions:

  1. In the introduction, it is asserted that “all amino acids share a common constant linear body and a variable side-chain”. Strictly, all α-amino acids (alpha-amino acids) share a common backbone. Natural proteins are made only of α-amino acids, but there do exist amino acids with longer or more exotic backbones that can be used in synthetic heteropolymer design. (Adding α ["alpha"] to the second and third sentences of the introduction should address this; a lengthy discussion of exotic amino acids would likely be beyond the scope of this work.)

Done.

  1. On line 37 of the introduction, “atomic forces that control the protein stability are represented as a decomposable energy function, defined as the sum of terms involving at most two bodies (atoms)”. That should be “at most two bodies (amino acids)”: energy functions often are not atom-level pairwise-decomposable. (In the Rosetta energy function, for example, there are one- and two-body torsional terms that are dependent on four atoms, but only on one or two amino acids. Older Rosetta energy functions, such as score12, also included a “pair” potential dependent only on the identities of amino acids in close proximity. This potential would be dependent on dozens of atoms but on only two amino acids.)

Change done. Indeed, one can understand that reasoning at the AA level allows for the encapsulation of more complex inter-atomic dependencies as long as they reside inside pairs of AAs’ dependencies.  Thanks for the clarification.

  1. On line 65 of the introduction, the term “energy gap” is a bit confusing, since this has other meanings in protein folding (usually, the difference in energy between a desired backbone conformation and all alternative conformations). I’d suggest “threshold”, the term used in the previous sentence.

 Done.

  1. It may be worth mentioning in the introduction that protein designers can now express and screen thousands of designs in yeast display libraries, as another reason that computer algorithms that produce diverse designs are more practical and useful than computer algorithms that produce only a single top-scoring design. Library design is a reason to use a diversity algorithm.

 A short sentence has been added.

  1. Patented sequences are given as one reason that a designer may wish to impose a requirement that designs have a minimum Hamming distance from a known sequence. Another reason that the authors may or may not wish to mention is antigenicity: there are known short peptide sequences (sub-sequences of a longer protein) that are recognized by the Major Histocompatibility Complexes (MHCs). If one wants to make a protein that could be used as a drug, it needs to evade the human immune system, which means that one wants sequences for which all sub-sequences have a Hamming distance greater than a given threshold from known antigenic sequences. Some computational methods exist for redesigning good designs to remove antigenic sub-sequences while trying to retain structure and function (e.g. King et al. (2014). Proc. Natl. Acad. Sci. USA 111(23):8577-8582), but means of preventing immunogenicity during design while preserving optimality guarantees would be very powerful. I leave it to the author’s discretion whether mentioning this would help the paper, or whether it would add unnecessary complexity.

It’s actually very interesting. A short sentence and the corresponding reference has been added. It’s very likely that specific automata-based constraints for such families of patterns could be encoded in a very concise form.

  1. Since the Rosetta ref2015 energy function is used, Rebecca Alford’s 2017 paper about this energy function (Alford et al. (2017) J. Chem. Theory Comput. 13(6):3031-3048) should probably be cited.

Definitely granted, reference added. The authors hope this energy function will be ultimately MIT licensed, as toulbar2 :-)

  1. On line 130, it is asserted that the computation of pairwise interaction energies is quadratic in protein length. This is actually a worst-case scaling. Since only positions close in space are considered to interact, and since each core residue has on average a constant number of neighbours, the scaling in practice tends to be O(N) with length instead of O(N^2). (The scaling is O(N^2) with number of rotamers per position, however).

Agreed again.  Distance cutoffs and their effect on complexity are now mentioned.

  1. On line 183, I think “solutions” should be possessive: “while maintaining the solutions’ joint costs unchanged”.

Done.

  1. In Definition 4, the authors state “Allowing both positive and negative threshold delta allows the DIST cost function to express either minimum or maximum diversity constraints.” It may be good to clarify whether it is possible to impose a minimum and a maximum for the same problem.

 We remained concise here. This ‘theoretical’ Dist constraint is specialized here either for minimum XOR maximum. One can use two constraints to express both at the same time. With a bit of extra code and the automata encoding, it would very easy to encode both constraints in a single automata. This is not currently done in the implementation: two constraints are still needed and this is usually absolutely tolerable.

  1. On line 282, the authors state, “We empirically confirmed this on very tiny instances.” It may be useful to state how tiny. At what point does the scaling make this approach impractical?

This has been tested on a problem (non CPD instance) with 20 variables and 6 values at most. Asking for four 15-diverse solutions took one day with the elegant DiverseSet encoding and 0.28” with the DiverseSeq one (roughly 300,000 times faster). This is now mentioned in the paper.

  1. On line 387, should “Figure 5” be “Figure 4”? (“Additional variables and ternary functions are represented in Figure 5.”)

Right. Corrected. Thanks!

  1. In the first paragraph of Section 8, it is stated that the tests were performed on a single core of a Xeon Gold 6140 CPU at 2.30 GHz. (The authors’ attention to detail is commendable.) Have the authors given any thought as to whether this algorithm can be efficiently parallelized? Implementing a parallel version is likely beyond the scope of this work, but a few words about the feasibility might be useful.

Systematic search algorithms are notoriously difficult to parallelize efficiently (topics of many CS research papers). But a parallel version of the HBFS search algorithm we developed and use is slowly being worked out in the lab. Note that this would increase the total CPU-time used (but save wall-clock time, although speedups are sublinear of course).

  1. In Figures 9 and 12, the colour key should be labelled. (Diversity or delta, I believe, if I am interpreting these figures correctly.)

 Done. Your interpretation is right.

  1. This is a stylistic point that I would also leave to the authors’ discretion: although it is intellectually honest and commendable of the authors to discuss current limitations and future directions in their concluding remarks, the last sentence of the paper ends on a bit of a pessimistic-sounding note: “...we have been unable to reach an empirically convincing result here,” (referring to a particular detail that might be explored further in the future, not to the overall work). It may be worthwhile to add a couple of additional sentences to end with the significance of the work in its present form. I do think this paper represents a significant contribution to protein design in its present form, and I wouldn’t want a reader to end with the wrongful impression that something more needs to be done in order for this work to be useful: it is unambiguously useful right now.

This has been removed and replaced by a mention of ongoing work by a PhD student.

Thanks again for all these interesting comments. It's always a pleasure to learn from a review.

Reviewer 2 Report

M. Ruffini, et al. detail the bases for automatic introduction of diversity constraints to insure a breadth of starting structures in the generation of engineered protein libraries.  Within the limitations of the approach, which notably treats all starting models with rigid polypeptide backbones, the authors show how their algorithm maximizes diversity in the ultimate energy minimized library.  They illustrate this with example cases from the Protein Data Bank.  Overall, the work is solid, the assumptions are made clear, and the message is very well communicated.

Page 1. The first word of the title is misspelled.  It should be "Guaranteed".

Author Response

Review 2:

  1. Ruffini, et al. detail the bases for automatic introduction of diversity constraints to insure a breadth of starting structures in the generation of engineered protein libraries.  Within the limitations of the approach, which notably treats all starting models with rigid polypeptide backbones, the authors show how their algorithm maximizes diversity in the ultimate energy minimized library.  They illustrate this with example cases from the Protein Data Bank.  Overall, the work is solid, the assumptions are made clear, and the message is very well communicated.

Page 1. The first word of the title is misspelled.  It should be "Guaranteed".

Thanks for the kind comments and for spotting this! This is corrected.

Reviewer 3 Report

General Comments:

Reduce the use of so-called throughout your paper.  Yes, they are called rotamer libraries.  You do not need the so-called

You never mention dynamics.  In your exposition of the problem, you expand on the fact that the rotamer packing problem exists and is huge in configurational space - but it's actually a much harder problem than that.  The fixed backbone is an assumption we use to reduce the problem space, but dynamics and the interaction of sequence and structure plays a very very large role.  This should be discussed somewhere. 

You compare your solution to toulbar 2, but I think adding a comparison to standard rosetta sequence design should be included.  This should be fairly striking, and would help in conferring the utility of the method.  Comparing both absolute energetics, time, and sequence recovery would be a really great additon to have in the results

Author Response

Review 3:

Reduce the use of so-called throughout your paper.  Yes, they are called rotamer libraries.  You do not need the so-called

Thanks for the suggestion. The amazingly numerous occurrences of “so-called” have been removed.

You never mention dynamics. In your exposition of the problem, you expand on the fact that the rotamer packing problem exists and is huge in configurational space - but it's actually a much harder problem than that.  The fixed backbone is an assumption we use to reduce the problem space, but dynamics and the interaction of sequence and structure plays a very very large role.  This should be discussed somewhere. 

The authors absolutely agree with the reviewer.  A subset of them actually contributed to a recent review paper on this topic:

Bouchiba, Younes, et al. "Molecular flexibility in computational protein design: an algorithmic perspective." Protein Engineering, Design and Selection 34 (2021)

This comment is also closely related with comments 1 and 2 of reviewer 1 and the corresponding paragraphs have been improved accordingly. Beyond “inverting deep learning based structure prediction approaches”, which are just starting to be explored, a partial answer to this lies in the various backbone flexibility-aware design approaches that have been proposed and are described in the review above. Most of them internally rely on iterated/relaxed/multi-state approaches that usually require solving a series (or a product of) rigid-backbone problems. In this case, they should be able to benefit from the encoding described here. We actually use this in combination with multi-state design.

You compare your solution to toulbar2, but I think adding a comparison to standard rosetta sequence design should be included.  This should be fairly striking, and would help in conferring the utility of the method.  Comparing both absolute energetics, time, and sequence recovery would be a really great additon to have in the results

We already use Rosetta’s rotamer library AND energy functions in all our molecular models, so the comparison would be a comparison between simulated annealing-based stochastic search as used in the fixbb protocol with systematic search as in toulbar2. Without the diversity constraint, this has been done in the following paper:

Simoncini, David, et al. "Guaranteed discrete energy optimization on large protein design problems." Journal of chemical theory and computation 11.12 (2015): 5980-5989.

Beyond this, it is extremely difficult to encode complex constraints as those used here in Monte Carlo algorithms. Complex constraints can easily break ergodicity. Ideally a “move” operator that preserves or controls the Hamming distance would be needed. This is a really demanding task. We hope the reviewer will understand this would require an amount of additional work that we would really prefer to avoid (not to mention the fact that this may prove to be infeasible or extremely hard).

Reviewer 4 Report

Ruffini et al. present a novel approach for producing diverse libraries for protein redesign using Cost Function Network algorithms with an automaton-based diversity constraint.

Overall the article is well written and thorough, with extensive background into how the Cost Function Network, automaton-based diversity constraints, and diversity metrics, such as Hamming distance, come into play to create diverse sequences for protein redesign. The background and methods are very well written, however, the results are confusing and difficult to follow. Most of the confusion stems from the representation and explanation of the results in figures 6 and 7 (and 10 and 11). Please add a more detailed figure caption and description of the figures in the results section.

Additional figure comments:

- The x-axis label does not provide enough explanation for what is represented in the plot.

- The y-axis is mislabeled in figures 7 and 11

- Figures 6 and 7 are almost identical. The expectation would be that NSSR should be higher than NSR because of the inclusion of non-identical, but similar sequences. Please add analysis or explanation for the results.

- Similar comment for figures 10 and 11. It is difficult to see a difference between the 4 figures.

Minor corrections:

- Title: Guaranteed is misspelled (missing an e)

- Page 3, last paragraph: van der Waals is misspelled (spelled as Walls)

- Page 16, first paragraph: reference to missing supplementary information

Author Response

Review 4:

Ruffini et al. present a novel approach for producing diverse libraries for protein redesign using Cost Function Network algorithms with an automaton-based diversity constraint.

Overall the article is well written and thorough, with extensive background into how the Cost Function Network, automaton-based diversity constraints, and diversity metrics, such as Hamming distance, come into play to create diverse sequences for protein redesign. The background and methods are very well written, however, the results are confusing and difficult to follow. Most of the confusion stems from the representation and explanation of the results in figures 6 and 7 (and 10 and 11). Please add a more detailed figure caption and description of the figures in the results section.

The captions have been detailed and improved.We hope this has now reached sufficient precision.

Additional figure comments:

- The x-axis label does not provide enough explanation for what is represented in the plot.

The X-axis is now systematically described in the captions. In all figures, the X-axis ranges over all tested backbones, sorted in increasing order of the presented measure.

- The y-axis is mislabeled in figures 7 and 11

Thanks for spotting this. This was actually the result of the inclusion of a wrong figure (that described a different similarity measure that we have not used in the paper for simplicity: the ratio of the alignment score of the designed sequence and the native with the alignment score of the native onto itself). This has been corrected, the labels and figures are now the suitable ones.

- Figures 6 and 7 are almost identical. The expectation would be that NSSR should be higher than NSR because of the inclusion of non-identical, but similar sequences. Please add analysis or explanation for the results.

The explanation for this is above. The NSSR is indeed higher than the NSR and the previous curves described a different measure (see above). This previous measure had been removed from the paper because it’s far less usual than NSSR.

- Similar comment for figures 10 and 11. It is difficult to see a difference between the 4 figures.

Again, same explanation. Corrections applied.

Minor corrections:

- Title: Guaranteed is misspelled (missing an e)

Thanks for spotting this. Corrected.

- Page 3, last paragraph: van der Waals is misspelled (spelled as Walls)

Thanks. Corrected.

- Page 16, first paragraph: reference to missing supplementary information

The supplementary information we are referring to is the Supplementary information provided with paper [46], not our paper. The sentence has been clarified. 

Thanks a lot for all the comments. We are extremely happy that the reviewer could spot the mislabelled figures!